

# Deriving the non-perturbative gravitational dual of quantum Liouville theory from BCFT operator algebra

**Lin Chen[1,2], Ling-Yan Hung[3,4]⋆, Yikun Jiang[5]† and Bing-Xin Lao[6,7]**

**1** School of Physics and Optoelectronics, South China University of Technology, Guangzhou 510641, China

**2** Institute of Fundamental Physics and Quantum Technology, & School of Physical Science and Technology, Ningbo University, Ningbo, Zhejiang 315211, China

**3** Yau Mathematical Sciences Center, Tsinghua University, Beijing 100084, China

**4** Yanqi Lake Beijing Institute of Mathematical Sciences and Applications (BIMSA), Huairou District, Beijing 101408, China

**5** Department of Physics, Northeastern University, Boston, MA 02115, USA

**6** École Normale Supérieure - PSL, 45 rue d'Ulm, F-75230 Paris cedex 05, France

**7** Sorbonne Université, CNRS, Laboratoire de Physique Théorique et Hautes Energies, LPTHE, F-75005 Paris, France

⋆ lyhung@tsinghua.edu.cn , † phys.yk.jiang@gmail.com

## Abstract

We demonstrate that, by utilizing the boundary conformal field theory (BCFT) operator algebra of the Liouville CFT, one can express its path-integral on any Riemann surface as a three dimensional path-integral with appropriate boundary conditions, generalising the recipe for rational CFTs [1–4]. This serves as a constructive method for deriving the *quantum* holographic dual of the CFT, which reduces to Einstein gravity in the large central charge limit. As a byproduct, the framework provides an explicit discrete state-sum of a 3D non-chiral topological theory constructed from quantum $6j$ symbols of $\mathcal{U}_q(sl(2,\mathbb{R}))$ with non-trivial boundary conditions, representing a long-sought non-perturbative discrete formulation of 3D pure gravity with negative cosmological constant, at least within a class of three manifolds. This constitutes the first example of an exact holographic tensor network that reproduces a known irrational CFT with a precise quantum gravitational interpretation.

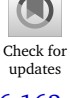

# 1   Introduction

To understand the AdS/CFT correspondence beyond the semi-classical limit, one crucial piece of information is the correct measure and collection of asymptotically AdS geometries that should be included in the path-integral of the AdS bulk to recover the CFT path-integral on any given manifolds. In this paper, we will illustrate with an example of a 2D irrational CFT that knowing the collection of conformal boundary conditions in the CFT would allow one to construct the 3D quantum geometrical sum explicitly. This is in line with the seminal series of results in 2D rational CFTs (RCFT) stating that the collection of boundary conditions completely determine the modular invariant spectrum of the full CFT [5]. Our construction adopts a framework based on some recent breakthroughs in triangulating 2D RCFTs in a controlled way [1–4]. Our strategy is that by working backwards from the triangulated 2D field theory, we will read off the 3D bulk quantum geometries that are being summed over, and along the way, the framework produces an exact tensor network description of both the bulk and the boundary CFT, allowing one to translate CFT degrees of freedom into AdS degrees of freedom readily. The quantum Liouville theory, namely as a unique solution of the modular bootstrap program [6–11], which is a rare occasion for an irrational CFT whose entire set of boundary conditions and structure coefficients are known, will be taken as our prime example to be studied in detail. As we are going to see, the boundary conditions would specify for us a very specific non-perturbative sum over AdS hyperbolic geometries. Admittedly, this sum would be rather unconventional for a "quantum gravitational theory", as important geometries, such as (spinning) BTZ black holes, could not play important roles due to the peculiar spectrum of the quantum Liouville theory. We do not claim that this is the expected version of a complete quantum gravitational theory, which is expected to include geometries such as BTZ black holes as dominant saddles in its ultimate UV complete quantum measure for its

path-integral. However, our work is a proof of principle that a UV complete quantum measure over geometries might not be unique and is a CFT dependent quantity. This should not be entirely surprising, as such a sum over geometries reproducing Liouville was anticipated to exist in prior studies of potential UV completions of 3D gravity, such as the Virasoro TQFT by *gauging a particular collection of topological lines* in the TQFT [12, 13].[1] However it is also known that the lines involved form a continuous spectrum, making it unclear how such gauging can be systematically implemented. This is a familiar problem in dealing with path-integrals of irrational TQFTs, where quantum measures often exhibit divergences absent in rational cases. This is the main reason that prevented prior attempts to discretise three dimensional gravitational theories rigorously on arbitrary manifolds, despite knowing that they are topological.[2] Our work essentially resolves this problem by generalising the new insights from the discrete formulation of RCFT discussed above. We recover an explicitly **convergent** quantum measure specifically for manifolds **with boundaries** specified by the CFT. These manifolds are central to the AdS/CFT correspondence [25–27], which inspired the current resolution that is otherwise not readily applicable to closed manifolds without boundaries, which have been the primary focus of much of the prior literature on the discrete formulation of quantum gravity.

In the framework developed in [1–4], it is shown that the path-integral of RCFTs can be discretised into a network of triangles. These triangles, which are conformally related to three point functions of boundary changing operators in the upper-half-plane, can be glued together using standard gluing methods in BCFT [28]. After they are glued together, the path-integral would be left with tiny holes at each vertex of the triangulation, with a conformal boundary condition fixed at each hole. The new ingredient is to make use of the so called "entanglement brane boundary condition" [1] – it is not a single local boundary condition, but a weighted sum of Cardy boundary conditions – to close these holes in a way explicitly preserving all the topological symmetries of the theory [2]. These boundaries are thus *shrinkable* and we will in the following call it instead the shrinkable boundary. The discretization essentially turns the CFT path-integral on arbitrary manifolds into a state-sum. Since these holes heal when they are sent to zero size, the resultant state-sum is independent of the initial triangulation. The methodology has been illustrated numerically in 2D minimal models to converge quickly even when only a few descendants are kept when gluing the triangles together [4].

In this paper, we would like to apply this discretisation procedure to Liouville theory, which is irrational. Fortunately, Liouville theory is one of the few examples of irrational CFTs where the complete set of conformal boundary conditions, boundary changing operators (BCO) and their corresponding structure coefficients have been obtained explicitly in the modular bootstrap program [9–11]. In this paper, we will supply also an appropriate shrinkable boundary, completing the task of triangulating the path-integral of an irrational CFT. There is a bonus to this triangulation. It makes the holographic relation between the *full CFT* path-integral and a topological field theory (TQFT) that is related to 3D gravity explicit. In the case of RCFTs, it is observed that the discretised partition function on a 2D manifold $\mathcal{M}$, can be written as a "strange correlator" $Z = \langle \Omega | \Psi \rangle$ [3, 4, 29–33]. Here, $|\Psi\rangle$ came from the three point BCO structure coefficients and the weighted sum prescribed in the shrinkable boundary. It is a ground state of the Levin-Wen string-net model [34], or equivalently, the 3D topological quantum field theory (TQFT) in Turaev-Viro (TV) formalism [35]. This 3D TQFT is the quantum double (i.e. square) of the Reshetikhin-Turaev TQFT [36, 37] associated to the Moore-Seiberg data of the RCFT [38]. Meanwhile, $\langle \Omega |$ is constructed from the tensor products of three point conformal blocks of the CFT. The strange correlator construction of RCFT path-integrals is an

---

[1]A more detailed analysis of the connection with the Virasoro TQFT [13] is provided in [14]. In the language of generalized symmetries [15–19], the "SymTFT" of Liouville theory is given by two copies of the Virasoro TQFT [13, 14].

[2]See for example literature on the Ponzano-Regge model [20–24]. For a review of the difficulty with convergence see also a recent discussion in [13].

explicit realisation of the holographic "sandwich" discussed in the non-invertible symmetry literature [16–19]. The non-invertible symmetry made explicit by the 3D TQFT is that associated to the topological line operators (i.e. Verlinde lines) in the RCFT [5,39–41]. In the case of Liouville theory, the discretised partition function can again be written as $Z_{\text{Liouville}} = \langle \Omega | \Psi_{\mathcal{U}_q(sl(2,\mathbb{R}))} \rangle$. The state $|\Psi_{\mathcal{U}_q(sl(2,\mathbb{R}))}\rangle$ came from the structure coefficients of the BCOs that take values proportional to quantum $6j$ symbols of some specific continuous representations[3] for the quantum group $\mathcal{U}_q(sl(2,\mathbb{R}))$ which is a non-compact quantum group. With a carefully chosen normalization of the Liouville operators, the structure of $|\Psi_{\mathcal{U}_q(sl(2,\mathbb{R}))}\rangle$ becomes explicitly analogous to that of RCFTs. It is tempting to interpret this state also as a ground state of a TV type TQFT associated to $\mathcal{U}_q(sl(2,\mathbb{R}))$, which is itself expressible as a state sum. As mentioned above, one difficulty with constructing a TV state sum for non-compact (quantum) group/irrational TQFT is that the sum (or integral) over representations could lead to divergences.[4] In this paper, rather than addressing the construction of irrational TQFTs on arbitrary 3-manifolds, we focus on the wave-function $\langle \Omega | \Psi_{\mathcal{U}_q(sl(2,\mathbb{R}))}\rangle$. The state $|\Psi_{\mathcal{U}_q(sl(2,\mathbb{R}))}\rangle$ is a path-integral over a 3 manifold with boundary on which the CFT lives, and is better behaved because the configuration of $6j$ symbols involved can be locally reduced to a situation where the orthogonality condition applies and the integral converges when combined to $\langle \Omega |$. The global wave-function is well-defined in scenarios where the Liouville path integral on the boundary manifold $\mathcal{M}$ is well-defined. In addition, we will show that the wavefunctions of the state $|\Psi_{\mathcal{U}_q(sl(2,\mathbb{R}))}\rangle$ indeed reduces to the 3D Einstein-Hilbert action evaluated in hyperbolic space in the semi-classical limit where the central charge $c$ is large.

Summarising, we establish three important results in this paper.

1. First, we introduce the shrinkable boundary of Liouville CFT, and express its path-integral $Z_{\text{Liouville}}(\mathcal{M})$ on any given 2D manifold $\mathcal{M}$ as a triangulated state sum, which in turn is a strange correlator i.e. $Z_{\text{Liouville}} = \langle \Omega | \Psi_{\mathcal{U}_q(sl(2,\mathbb{R}))}\rangle$. **The state $|\Psi_{\mathcal{U}_q(sl(2,\mathbb{R}))}\rangle$ involved is locally finite, and globally depends on the number of punctures and boundaries of the boundary manifold $\mathcal{M}$.** The remaining divergences in $\langle \Omega | \Psi_{\mathcal{U}_q(sl(2,\mathbb{R}))}\rangle$ arise when the holes in the triangulation are reduced to zero size, originating from the Casimir energy. These divergences can be systematically regulated and divided out.

2. Second, $|\Psi_{\mathcal{U}_q(sl(2,\mathbb{R}))}\rangle$ can be interpreted as gluing of hyperbolic tetrahedra $T_i$. The BCO and boundary conditions play the role of parametrising the (geodesic) edge-lengths of these hyperbolic tetrahedra, determining the shape and volume of each tetrahedron. Each $T_i$ contributes to a quantum $6j$ symbol which can be identified with the gravitational on-shell action with corner term on the hyperbolic tetrahedron, in the large $c$ limit of Liouville theory. **In the large $c$ limit, the Liouville partition function reduces to a specific sum over hyperbolic geometries $H$, whose boundary is $\mathcal{M}$, weighted by the AdS$_3$ gravitational Einstein-Hilbert action.** The shrinkable boundary ensures that all corner terms drop out precisely when the gluing of tetrahedra is smooth.

3. The triangulation in $|\Psi_{\mathcal{U}_q(sl(2,\mathbb{R}))}\rangle$ can be changed, not completely arbitrarily, but up to the pentagon relation (reviewed in the appendix) and orthogonality relation (9) satisfied by the quantum $6j$ symbols. The edges of the hyperbolic tetrahedron are geodesics in 3D hyperbolic space, which form the basic degrees of freedom of the 3D theory. Note that while the 3D bulk theory is topological, the boundary condition is not. Exactly as in the continuous case, the boundary gives meaning to radial distances in the bulk. To access

---

[3]These representations are representations of the 'modular double' of $\mathcal{U}_q(sl(2,\mathbb{R}))$ [8, 42], where $q = e^{i\pi b^2}$. They have the special property that they are also representations of $\mathcal{U}_{\tilde{q}}(sl(2,\mathbb{R})), \tilde{q} = e^{i\pi/b^2}$.

[4]The Teichmuller TQFT constructed by Andersonn-Kashaev [43,44] is however a rigorous discrete state-sum and it is believed to be related to AdS$_3$ gravity, although its precise relation is yet to be understood.

CFT physics at different scales, one can recursively coarse-grain the triangulation using the steps taken in [3]. These coarse-graining steps produce a flow in the boundary conditions, and allows one to read off the radial direction of the bulk. **This thus generates a natural, exact and well-behaved holographic tensor network that is constructed from a state-sum of hyperbolic tetrahedra.** The edges in the holographic network are geodesics connecting vertices separated by different distances at the boundary $\mathcal{M}$. A geodesic network is thus recovered in the bulk theory.

We would like to take a moment to provide some general context for the holographic tensor network literature. Since Swingle's seminal work in 2009, which conjectured that tensor networks could serve as the correct framework to capture the microscopic relationship between CFT and AdS degrees of freedom [45], there has been a surge of interest in constructing tensor networks capable of recovering both the CFT and its holographic dual. However, for over a decade, it has proven exceedingly difficult to construct *any* tensor network—numerical or analytical, discrete or continuous—for *any* CFT, whether free, rational, or irrational, in any dimension, that meets some basic expectations. These expectations include the rigorous recovery of the CFT path-integral or wave function, the *emergence* of a concrete bulk theory with a precise action (in fact the Einstein Hilbert action, should the dual be gravitational at all), and graph independence or covariance, among other criteria. These challenges have cast doubt on whether tensor networks can ever be taken seriously as a quantitative framework for understanding the AdS/CFT correspondence. One major difficulty lies in the fact that constraining the choice of tensors or graphs solely through requirements on entanglement entropy is too weak to sufficiently narrow down the space of possible tensors. What makes the current construction stand out is its use of sufficiently many generalized symmetries, in particular non-invertible symmetries [5,15,38–41,46–48], which successfully constrain the space of tensors to recover the exact CFT. Remarkably, the emergence of geometry is naturally handled by the CFT itself when it is rigorously reproduced. To date, this is the first and only analytic, graph-independent holographic tensor network where both the CFT and its holographic dual are rigorously under control. Furthermore, as envisioned by Swingle, it indeed provides a microscopic map between the degrees of freedom in the CFT and its geometrical dual.

Admittedly, Liouville CFT is not a typical holographic CFT, as its holographic dual features a highly unconventional quantum measure as a "gravity" theory [49–52]. Despite its seemingly "simple" nature, constructing such a precise microscopic framework for Liouville CFT requires brand new strategies and techniques in generalized symmetries that have only recently become available, to show that such a precise microscopic construction is in fact possible. We hope our construction introduces a framework that is perhaps anticipating the next breakthrough in the rigorous reconstruction of more interesting interacting CFTs and their duals.

## 2    Shrinkable boundary and discretising Liouville theory

Essential details of Liouville CFT are reviewed in the appendices. Similar to the case of RCFTs, the basic building blocks of triangulating the Liouville CFT partition function are the three point functions of the boundary changing operators. These boundary structure coefficients have been obtained via modular bootstrap [11]. For later convenience, we will pick a carefully chosen different normalisation. The details of our convention and definitions can be found in Appendix A.

Each primary boundary changing operator is labeled by three parameters as

$$\Phi_\alpha^{\sigma_1\sigma_2}(x).\tag{1}$$

The boundary conditions are Cardy boundary conditions parametrised by a parameter which

are labeled $\sigma_1$ and $\sigma_2$ above. In Liouville theory, $\sigma_{1,2}$ and $\alpha$ can all be parametrised as $\frac{Q}{2}+i\mathbb{R}_{\geq 0}$ [9, 10], where $Q = b + 1/b$ is a parameter in the theory that controls the central charge, i.e. $c = 1 + 6Q^2$. The conformal weight of the primary operator is given by $\Delta = \alpha(Q - \alpha)$. For later convenience, we introduce the following notations

$$\sigma \equiv Q/2 + iP_\sigma\,, \qquad \alpha \equiv Q/2 + iP_\alpha\,, \tag{2}$$

with $P_\sigma, P_\alpha \geq 0$. Unless where they appear in explicit functions, we might use $\sigma, \alpha$ and $P_\sigma, P_\alpha$ interchangeably as labels of the boundary conditions and conformal primaries.

In our normalisation the boundary structure coefficients are defined by three point correlation functions on the upper half plane as[5]

$$\langle \Phi_{\alpha_1}^{\sigma_1\sigma_2}(x_1)\Phi_{\alpha_2}^{\sigma_2\sigma_3}(x_2)\Phi_{\alpha_3}^{\sigma_3\sigma_1}(x_3)\rangle = \frac{C_{\alpha_1,\alpha_2,\alpha_3}^{\sigma_3,\sigma_1,\sigma_2}}{|x_{21}|^{\Delta_1+\Delta_2-\Delta_3}|x_{32}|^{\Delta_2+\Delta_3-\Delta_1}|x_{31}|^{\Delta_3+\Delta_1-\Delta_2}}\,, \tag{3}$$

and take the following form:

$$C_{\alpha_1,\alpha_2,\alpha_3}^{\sigma_3,\sigma_1,\sigma_2} = \frac{\left(\mu(P_{\alpha_1})\mu(P_{\alpha_2})\mu(P_{\alpha_3})\right)^{1/4}}{2^{3/8}\sqrt{\gamma_0}\,\Gamma_b(Q)} \times \sqrt{C(\alpha_1,\alpha_2,\alpha_3)}\begin{Bmatrix}\alpha_1 & \alpha_2 & \alpha_3 \\ \sigma_3 & \sigma_1 & \sigma_2\end{Bmatrix}_b\,, \tag{4}$$

where $\mu(P_\alpha)$ is the Plancherel measure that we will introduce in a moment. $\gamma_0$ and $\Gamma_b(Q)$ are constants whose expressions can be found in Appendix A. $C(\alpha_1, \alpha_2, \alpha_3)$ is the closed three point structure coefficient (A.7) in our normalization. $\begin{Bmatrix}\alpha_1 & \alpha_2 & \alpha_3 \\ \sigma_3 & \sigma_1 & \sigma_2\end{Bmatrix}_b$ is the $6j$ symbol for some representations of $\mathcal{U}_q(sl(2,\mathbb{R}))$ [11] as in (A.12), with

$$q = e^{i\pi b^2}\,. \tag{5}$$

The chosen normalization ensures that the boundary Liouville operators are invariant under the Liouville reflection $P \to -P$ and renders the form of the BCFT OPE coefficients *exactly* equivalent to those of rational CFTs [3, 53], expressed in terms of quantum $6j$ symbols.

Under this choice of normalisation, boundary two point functions are given by

$$\langle \Phi_{\alpha_1}^{\sigma_1\sigma_2}(0)\Phi_{\alpha_2}^{\sigma_2\sigma_1}(1)\rangle = 2\pi\delta(P_{\alpha_1} - P_{\alpha_2})\,. \tag{6}$$

The Liouville partition function will be constructed by gluing the boundary three point functions together. The method is identical to that described in [4]. We first map the three point function computed on the upper half plane to these triangles, which we denote by $\gamma_{I_1 I_2 I_3}^{\alpha_1\alpha_2\alpha_3}(\epsilon)$, by a conformal transformation. The indices $I_i$ denotes the level number of the operator as a descendent in the Virasoro family $\alpha_i$. For a fixed shape of the triangle, the conformal map from the upper-half plane is fixed. We picked one in [4] that produces right-angled isosceles triangles with their vertices slightly chipped. The chipped corner is a short conformal boundary that has length of order $\epsilon$. The detail of the conformal map is not very important for our purpose here and we refer interested readers to [4] for details. The edges of the triangles are labeled by Virasoro representation $\alpha$, and also the level number in the corresponding representation. Each of the chipped vertices is labeled by conformal boundary conditions $\sigma$. These triangles are then glued to others sharing an edge by summing over all the representations and their corresponding Virasoro descendants on the shared edge with matching boundary conditions at the two vertices the edge connects to. This is portrayed in the figure 1 and figure 2.

---

[5]Our choice of ordering of indices in the structure coefficients are deliberately matching the form of the quantum $6j$ symbol. This is different from the more commonly adopted notation found for example in [11].

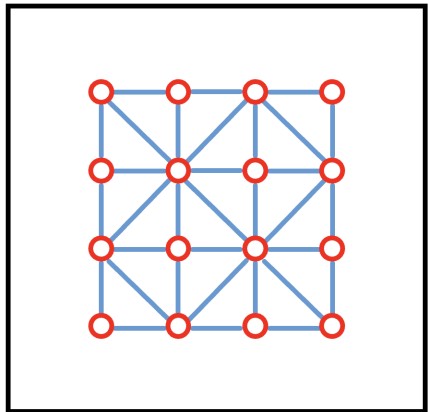

Figure 1: One choice of tiling of the plane by triangles whose angles are slightly chipped.

In the case of diagonal RCFTs, each of the corner label $a$ is summed with a weight $S_{\mathbf{1}a}\sqrt{S_{\mathbf{11}}}$, where $S_{ij}$ are components of the modular transformation matrix of the chiral conformal primary characters on a torus, and $\mathbf{1}$ labels the identity sector. This sum of Cardy states is proved to reproduce exactly the vacuum Ishibashi state in the dual closed channel:

$$|\mathbf{1}\rangle\rangle = \sum_a S_{\mathbf{1}a}\sqrt{S_{\mathbf{11}}}|a\rangle_{\text{Cardy}}. \tag{7}$$

In the small hole limit, the vacuum Ishibashi state converges quickly to the vacuum state, closing the hole while preserving all the topological symmetries [2]. In the case of Liouville theory, the vacuum state is not in the physical spectrum. A non-physical degenerate state with primary label given by $P_{\mathbf{1}} = iQ/2$ (i.e. $\sigma_{\mathbf{1}} = 0$) and conformal dimension 0 is the closest analogue of the vacuum state. We will simply call it the vacuum state in the following. Therefore, we would like to find a weighted integral of the Cardy boundary conditions so that it produces the Ishibashi state constructed from this vacuum family. The natural guess of the weight is $S_{\mathbf{1}\alpha}$ in Liouville theory, which is also known as the Plancherel measure $\mu(P_\alpha)$. A related proposal for holographic CFTs assumed to contain the vacuum state features also the Plancherel measure [54, 55]. It is given by[6]

$$\mu(P) = 4\sqrt{2}\sinh(2\pi Pb)\sinh\left(\frac{2\pi P}{b}\right). \tag{8}$$

There is an independent condition that suggests that this is the correct weight in the sum over boundary conditions. Consider the open structure coefficients of Liouville theory. They are proportional to the $6j$ symbols of $\mathcal{U}_q(sl(2,\mathbb{R}))$. Importantly, they satisfy the pentagon equation and orthogonality condition. They can be represented graphically as in figure 3.

The pentagon equation is reviewed in the appendix, and the orthogonality condition is reproduced here.

$$\int_0^\infty dP_{\sigma_3}\mu(P_{\sigma_3})\begin{Bmatrix} \alpha_2 & \alpha_3 & \alpha_1 \\ \sigma_1 & \sigma_2 & \sigma_3 \end{Bmatrix}_b \begin{Bmatrix} \alpha_2 & \alpha_3 & \alpha_1' \\ \sigma_1 & \sigma_2 & \sigma_3 \end{Bmatrix}_b = \frac{2\delta(P_{\alpha_1} - P_{\alpha_1'})}{\mu(P_{\alpha_1})}. \tag{9}$$

This equation can be interpreted as the structure coefficients on a strip with a hole reducing to the structure coefficients on a strip after the conformal boundary condition $\sigma_3$ is integrated with the Plancherel measure.(See the bottom picture of figure 3.)

---

[6]See reviews such as [56] and references therein.

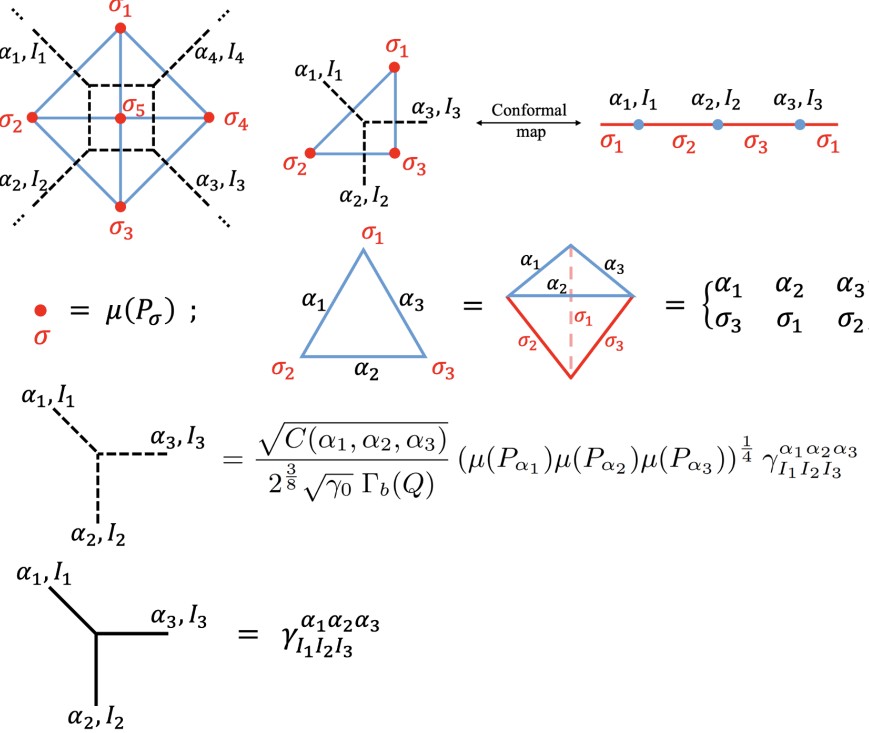

Figure 2: The top left picture is a particular choice of triangulation of the CFT partition function. These triangles and their dual graphs correspond to open structure coefficients and conformal blocks respectively. The dashed vertex corresponds to 3-point block in Racah gauge absorbing the Plancherel measure as normalisation. The solid line vertex corresponds to 3-point block in block gauge.

This is the first important indication to us that the Plancherel measure is the correct weight. To check that the prescription is correct, we make use of the crossing symmetry of the CFT. For a given triangulation, consider its dual graph. For fixed edge labels, the dual graph represents the conformal blocks in specific intermediate fusion channels. Crossing relations of the conformal blocks is illustrated in figure 4.

Writing the structure coefficients as $6j$ symbols (4), and then using the pentagon identity and orthogonality as shown in figure 3, we can show that the combination with conformal blocks is crossing symmetric, as illustrated in figure 10. This allows one to reduce any loop in the dual graph to one containing a loop with exactly two external legs. One example of such a reduction is illustrated in figure 5.

Therefore we need only to show that the hole in the interior of the bubble inside the shaded region in figure 5 under the integral of the conformal boundary weighted by the Plancherel measure produces the $P_1$ Ishibashi state in the closed channel.

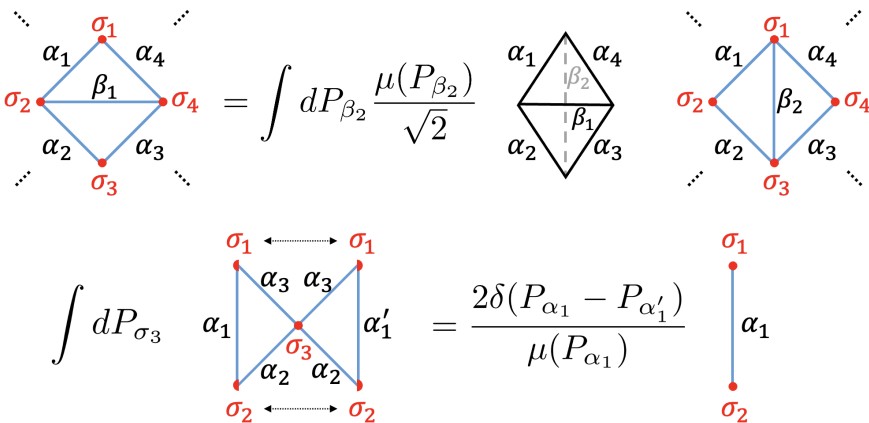

Figure 3: The triangles with contracted red dots describe the wave-function $|\Psi\rangle$. Each of these triangles are $6j$ symbols, and they satisfy the pentagon relation (top) and the orthogonality condition (bottom) which serves to connect wave-functions of different triangulations via local moves. In equation form, the pentagon relation is given by (A.14), while the orthogonality condition is expressed in (9).

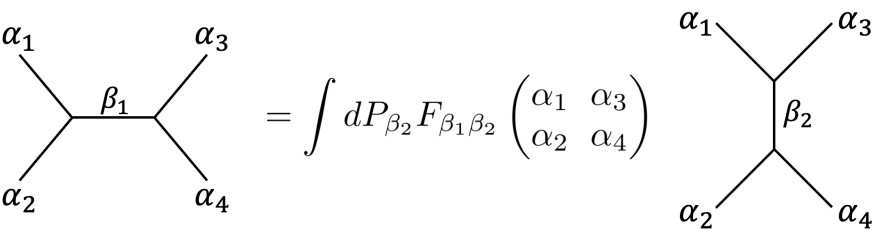

Figure 4: Crossing relations of the conformal blocks.

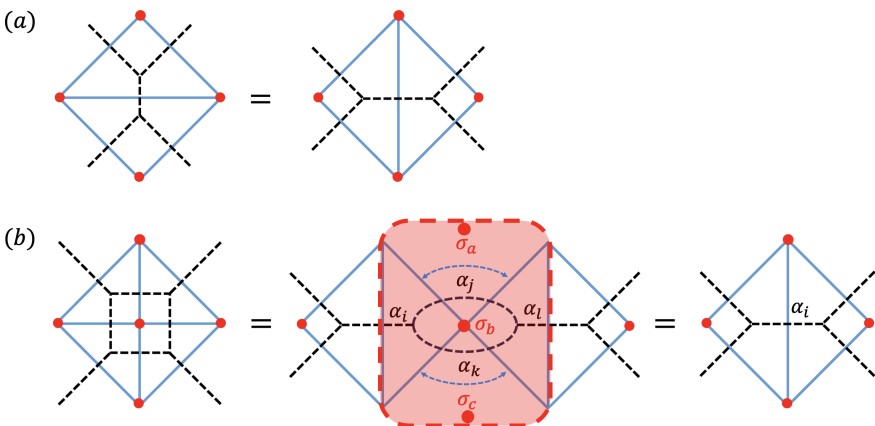

Figure 5: (a) Graphical representation of the associativity for boundary four point conformal blocks. (b) Example of reducing a graph with a loop to a graph containing a bubble with 2 legs (area circled by dashed lines and shaded red) via crossing relations. The blue dashed arrows indicate that those pairs of edges actually are identified. We label the edges to match with equation (10).

Substituting our choice of weighted sum of boundary conditions at the vertex, we obtain[7]

$$
\int_0^\infty dP_{\alpha_j} dP_{\alpha_k} dP_{\sigma_b} \mu(P_{\sigma_b}) C^{\sigma_b,\sigma_c,\sigma_a}_{\alpha_i,\alpha_j,\alpha_k} C^{\sigma_c,\sigma_b,\sigma_a}_{\alpha_j,\alpha_l,\alpha_k}
$$
[diagram with $\alpha_i$, $\alpha_j$, $\tau$, $\alpha_l$, $\alpha_k$]
$$
= 2^{-\frac{3}{4}} \left( \frac{1}{\sqrt{\gamma_0}\Gamma_b(Q)} \right)^2 \int_0^\infty dP_{\alpha_j} dP_{\alpha_k} dP_{\sigma_b} \mu(P_{\sigma_b})
$$

$$
\times [\mu(P_{\alpha_i})\mu(P_{\alpha_l})]^{\frac{1}{4}} \sqrt{\mu(P_{\alpha_j})\mu(P_{\alpha_k}) C(\alpha_i,\alpha_j,\alpha_k) C(\alpha_l,\alpha_j,\alpha_k)}
$$

$$
\times \begin{Bmatrix} \alpha_k & \alpha_j & \alpha_i \\ \sigma_a & \sigma_c & \sigma_b \end{Bmatrix}_b \begin{Bmatrix} \alpha_k & \alpha_j & \alpha_l \\ \sigma_a & \sigma_c & \sigma_b \end{Bmatrix}_b \quad
$$
[diagram with $\alpha_i$, $\alpha_j$, $\tau$, $\alpha_l$, $\alpha_k$]

$$
= \delta(P_{\alpha_i} - P_{\alpha_l}) 2^{\frac{1}{4}} \left( \frac{1}{\sqrt{\gamma_0}\Gamma_b(Q)} \right)^2 \int_0^\infty dP_{\alpha_j} dP_{\alpha_k}
$$

$$
\times \sqrt{\frac{\mu(P_{\alpha_j})\mu(P_{\alpha_k})}{\mu(P_{\alpha_i})}} C(\alpha_i,\alpha_j,\alpha_k) \quad
$$
[diagram with $\alpha_i$, $\alpha_j$, $\tau$, $\alpha_l$, $\alpha_k$]

$$
= \delta(P_{\alpha_i} - P_{\alpha_l}) \frac{2^{\frac{1}{4}}}{c_b} \left( \frac{1}{\sqrt{\gamma_0}\Gamma_b(Q)} \right)^2 \int_0^\infty dP_{\alpha_j} dP_{\alpha_k} \mu(P_{\alpha_k}) F_{\mathbf{1}\alpha_j}
$$

$$
\times \begin{pmatrix} \alpha_i & \alpha_k \\ \alpha_i & \alpha_k \end{pmatrix} \quad
$$
[diagram with $\alpha_i$, $\alpha_j$, $\tau$, $\alpha_l$, $\alpha_k$]

$$
= \delta(P_{\alpha_i} - P_{\alpha_l}) \frac{2^{\frac{1}{4}}}{c_b} \left( \frac{1}{\sqrt{\gamma_0}\Gamma_b(Q)} \right)^2 \int_0^\infty dP_{\alpha_k}
$$

$$
\times \mu(P_{\alpha_k}) \quad
$$
[diagram with $\alpha_i$, $\alpha_l$, $\mathbf{1}$, $\alpha_k$, $\tau$]

$$
= \delta(P_{\alpha_i} - P_{\alpha_l}) \frac{2^{\frac{1}{4}}}{c_b} \left( \frac{1}{\sqrt{\gamma_0}\Gamma_b(Q)} \right)^2 \quad
$$
[diagram with $\alpha_i$, $\alpha_l$, $\mathbf{1}$, $\mathbf{1}$, $\mathbf{1}$, $-1/\tau$],

$$(10)$$

where in the first equation, we plug in the expressions for the boundary three point functions (4). In the second equation, we use the orthogonality condition (9) for the $6j$ symbols. Then in the third equation, we rewrite the bulk OPE coefficient in terms of the Virasoro fusion kernel related to the identity module [57] using (A.29), and changes the conformal block into its dual channel with identity module propagation. In the last equation, we use the fact that the Plancherel measure $\mu(P_{\alpha_k})$ is equal to the modular S-matrix $S_{\mathbf{1}\alpha_k}$.

So we show that the hole indeed reduces to the $P_{\mathbf{1}}$ Ishibashi state we denote as $|\mathbf{1}\rangle\rangle$ in the closed string channel exactly as in the case of the RCFTs [1, 2, 4], despite the fact that this primary is not even in the physical spectrum of Liouville theory! The identity module pops out from the relation between bulk OPE coefficients and the Virasoro fusion kernel [57]. This weighted sum of boundary conditions produce the shrinkable boundary of Liouville theory. In the above equation $\tau$ is parametrising the circumference of the hole i.e. $\tau = \frac{i\epsilon}{2\pi}$.[8] In the last line of (10), the vacuum Ishibashi state is evolved by the closed CFT Hamiltonian by a distance

---

[7]Here, we choose to do the computation in the standard 'block gauge' instead of 'Racah gauge' as defined below in (15). The two choices of gauge would give the same final answer, but the block gauge has the convergence property manifest, as in noncompact CFTs, we need to divide and multiply by infinite factors for the structure coefficients and conformal blocks respectively in Racah gauge.

[8]To be precise, suppose the actual radius of the hole on the flat plane is $R$, and the lattice spacing is $L$, then $-\frac{\pi}{\epsilon} = -\ln(\frac{L}{R})$ [28]. And the $\epsilon \to 0$ limit corresponds to $R \to 0$.

$-1/\tau$. The Hamiltonian in the closed channel is given by $H_{closed} = \hat{D} - c/12$, where $\hat{D} = \hat{L}_0 + \hat{\bar{L}}_0$ is the dilatation operator. Putting them together, the bubble in the last line contributes

$$\cdots e^{-\frac{2\pi}{\epsilon}(\hat{D}-c/12)}|\mathbf{1}\rangle\rangle = e^{\frac{\pi c}{6\epsilon}} \cdots e^{-\frac{2\pi\hat{D}}{\epsilon}}|\mathbf{1}\rangle\rangle, \tag{11}$$

where $\cdots$ denote the rest of the state sum involving other parts of the lattice. Since the spectrum of $\hat{D}$ is non-negative and only the vacuum and its descendants can contribute, and they contribute with a discrete spectrum, the divergence comes from $e^{\frac{\pi c}{6\epsilon}}$ which is a universal contribution from every hole, and diverges in the limit $\epsilon \to 0$. Therefore to recover $Z_{\text{Liouville}}$, we have to divide every hole by a factor of $e^{\frac{\pi c}{6\epsilon}}$ before taking the limit $\epsilon \to 0$. Factors of quantum dimension $\mathcal{D} = \sqrt{\sum_i d_i^2}$ introduced at every vertex in the usual TV state sum is not needed explicitly to reproduce the Liouville path-integral. This by-passes the issue that this factor is infinity for a generic irrational TQFT and thus ill-defined. Note that the second equality follows from the orthogonality condition (9). This is what ensures that the wave-function that we will define below in (16) is locally finite. We are only left with divergence involving the small size of the hole, which is a UV divergence that can be readily regulated and subtracted. The procedure of subtracting this small hole divergence is the same as what is adopted in the RCFT case. As we will see, this is in fact similar to the usual IR cutoff in AdS/CFT near the asymptotic boundary.

## 3  Sum over geometries from strange correlator

The Liouville path-integral above is built out of triangles which are conformally related to three point correlation functions of three BCO on the upper half plane. It can be expressed as

$$\mathcal{T}^{\sigma_1\sigma_2\sigma_3}_{(\alpha_1,I_1)(\alpha_2,I_2)(\alpha_3,I_3)}(\triangle) = C^{\sigma_3\sigma_1\sigma_2}_{\alpha_1\alpha_2\alpha_3}\gamma^{\alpha_1\alpha_2\alpha_3}_{I_1I_2I_3}(\epsilon). \tag{12}$$

As described earlier, the conformal map to the triangle completely fixes the form of $\gamma^{\alpha_1\alpha_2\alpha_3}_{I_1I_2I_3}(\epsilon)$, which depends explicitly on the size of the holes $\epsilon$. The three point functions of boundary changing primary operators in the upper half plane is reproduced in (A.22). The structure coefficients are fixed after fixing normalisation of two point function as (6).

Therefore, the path-integral can be written as

$$Z_{\text{Liouville}} = \lim_{\epsilon\to 0} \prod_v \int_0^\infty dP_{\sigma_a} (\mathcal{N}(\epsilon)\mu(P_{\sigma_a})) \times \prod_e \int_0^\infty dP_{\alpha_i} \sum_{\{I_i\}}\prod_\triangle \mathcal{T}^{\sigma_a\sigma_b\sigma_c}_{(\alpha_i,I_i)(\alpha_j,I_j)(\alpha_k,I_k)}(\triangle),$$
$$\mathcal{N}(\epsilon) = 2^{-1/4}c_b(\sqrt{\gamma_0}\Gamma_b(Q))^2 e^{-\frac{\pi c}{6\epsilon}}, \tag{13}$$

where $v$ denotes the vertices, labeled by $a, b, c$, and $e$ represents the edges, labeled by $i, j, k$. This expression can be written as

$$Z_{\text{Liouville}} = \lim_{\epsilon\to 0}\langle\Omega|\Psi_{\mathcal{U}_q(sl(2,\mathbb{R}))}\rangle. \tag{14}$$

In direct analogy to the case of RCFTs [3, 4], it is more enlightening to re-scale the 3-point block $\gamma^{\alpha_1\alpha_2\alpha_3}_{I_1I_2I_3}(\epsilon)$ from the standard normalisation we have adopted here to the so-called Racah gauge

$$\tilde{\gamma}^{\alpha_1\alpha_2\alpha_3}_{I_1I_2I_3}(\epsilon) = \frac{\sqrt{C(\alpha_1,\alpha_2,\alpha_3)}}{2^{3/8}\sqrt{\gamma_0}\Gamma_b(Q)}\gamma^{\alpha_1\alpha_2\alpha_3}_{I_1I_2I_3}(\epsilon). \tag{15}$$

In this gauge the structure coefficients would change accordingly to keep the 2-pt and 3-pt function invariant. They would take then the standard form of quantum $6j$ symbols with

explicit tetrahedral symmetry. Then we can write

$$|\Psi_{\mathcal{U}_q(sl(2,\mathbb{R}))}\rangle = \prod_i \int_0^\infty dP_{\alpha_i} \sqrt{\mu(P_{\alpha_i})} \prod_v \int_0^\infty dP_{\sigma_a} (\mathcal{N}(\epsilon)\mu(P_{\sigma_a})) \prod_\triangle \begin{Bmatrix} \alpha_i & \alpha_j & \alpha_k \\ \sigma_c & \sigma_a & \sigma_b \end{Bmatrix}_b |\{\alpha_i\}\rangle, \tag{16}$$

and

$$\langle\Omega| = \prod_i \int_0^\infty dP_{\alpha_i} \sum_{\{I_i\}} \langle\{\alpha_i\}| \prod_\triangle \left(\tilde{\gamma}_{I_iI_jI_k}^{\alpha_i\alpha_j\alpha_k}(\epsilon)\right). \tag{17}$$

Here, $\prod_\triangle |\{\alpha_i\}\rangle$ are basis states on a 2D surface for the 3D TV TQFT, and they are defined in an analogous way to the Levin-Wen model [34] for rational TQFT. The 2D surface on which the state is defined has been triangulated, and each edge is coloured by the parameter $\alpha_i$. The basis on each edge with different colors are orthogonal to each other, although when two states on the same surface are constructed with different triangulations they are generically not linearly independent, and they are related by a linear map constructed using quantum $6j$ symbols, basically using the rules already detailed in figure 3. We observe that for each fixed boundary surface primary label basis state $|\{\alpha_i\}\rangle$ in $|\Psi_{\mathcal{U}_q(sl(2,\mathbb{R}))}\rangle$, the probability amplitude reduces to the exponent of the gravity action evaluated on-shell on a hyperbolic space whose boundary is the surface $\mathcal{M}$ defining the CFT.

To see that, first each $6j$ symbol is associated with a tetrahedron. One face of the tetrahedron carries the three edges labeled by the primaries $P_{\alpha_i}$ and this surface is where the CFT conformal block is attached to. The edges carrying the boundary condition labels $P_\sigma$ point in a direction orthogonal to the surface on which the CFT lives. This is illustrated in figure 6.

The classical limit when the parameter $b \to 0$ of quantum $6j$ symbols has been considered [58]. This corresponds to $Q \to \infty$ and so the central charge $c = 1 + 6Q^2$ also approaches infinity. The volume conjecture implies that [43,58,59]

$$\lim_{b\to 0} \begin{Bmatrix} \frac{\theta_1}{2\pi b} & \frac{\theta_2}{2\pi b} & \frac{\theta_3}{2\pi b} \\ \frac{\theta_4}{2\pi b} & \frac{\theta_5}{2\pi b} & \frac{\theta_6}{2\pi b} \end{Bmatrix}_b = \exp\left(-\frac{V(\theta_1,\theta_2,\theta_3,\theta_4,\theta_5,\theta_6)}{\pi b^2}\right), \tag{18}$$

where $V(\theta_1,\theta_2,\theta_3,\theta_4,\theta_5,\theta_6)$ is the volume of a hyperbolic tetrahedron[9] with dihedral angles $0 \le \theta_i \le \pi$. This corresponds to $\alpha_i = \theta_i/(2\pi b)$, and thus $P_i = iQ/2 - i\theta_i/(2\pi b)$ i.e. $P_i$ is purely imaginary and in some very limited range. This seems very far from where we are computing the $6j$ symbols for Liouville theory, in which $P_i$ is real and positive, and that they can take very large values. Therefore, we would like to interpret $P_i$ as parametrising the geodesic distance somehow, rather than dihedral angles of a tetrahedron. To do so, we note that there is a very interesting formula that expresses the volume of a hyperbolic tetrahedron in terms of its geodesic edge lengths rather than dihedral angles [64]:

$$\text{Vol}(T) = V(\theta_1,\theta_2,\theta_3,\theta_4,\theta_5,\theta_6) = V_l - \sum_i l_i \partial_{l_i} V_l, \tag{19}$$

$$V_l \equiv V(\pi - il_4, \pi - il_5, \pi - il_6, \pi - il_1, \pi - il_2, \pi - il_3), \tag{20}$$

where $V(\theta_1,\theta_2,\theta_3,\theta_4,\theta_5,\theta_6)$ is the volume $\text{Vol}(T)$ of the hyperbolic tetrahedron $T$ as a function of its diheral angles, and $V(\pi - il_4, \pi - il_5, \pi - il_6, \pi - il_1, \pi - il_2, \pi - il_3)$ is the same function with complex input variables. Here, $l_i$ is the geodesic length of the same edge where the dihedral angle $\theta_i$ is defined in the given tetrahedron. In fact [64],

$$\theta_i = 2\partial_{l_i} V_l, \qquad \text{mod } 2\pi. \tag{21}$$

---

[9]More accurately, it is discussed in detail that the tetrahedron is actually a "truncated hyperideal tetrahedron" [60–63]. This slightly changes the corner region of the tetrahedra where they are regulated, but does not alter most of the discussion in the current paper.

We note that $V(\pi - il_4, \pi - il_5, \pi - il_6, \pi - il_1, \pi - il_2, \pi - il_3)$ itself is not the volume of a tetrahedron. Now it is very tempting to take (18), and analytically continue it to complex dihedral angles i.e. replace $\theta \to \pi - il$, for all $l > 0$. The left hand side of (18) would correspond to taking $\alpha_i \equiv Q/2 + iP_i = Q/2 - il_i/(2\pi b)$. Since the $6j$ symbol is symmetric under $P_i \to -P_i$, we take the analytic continuation of (18)[10]

$$\lim_{b \to 0} \left\{ \begin{matrix} \frac{Q}{2} + i\frac{l_4}{2\pi b} & \frac{Q}{2} + i\frac{l_5}{2\pi b} & \frac{Q}{2} + i\frac{l_6}{2\pi b} \\ \frac{Q}{2} + i\frac{l_1}{2\pi b} & \frac{Q}{2} + i\frac{l_2}{2\pi b} & \frac{Q}{2} + i\frac{l_3}{2\pi b} \end{matrix} \right\}_b = \exp\left( -\frac{V(\pi - il_4, \pi - il_5, \pi - il_6, \pi - il_1, \pi - il_2, \pi - il_3)}{\pi b^2} \right)$$

$$= \exp\left( -\frac{\mathrm{Vol}(T\{l_i\}) + \sum_i l_i \theta_i / 2}{\pi b^2} \right), \tag{22}$$

where $T(\{l_i\})$ is the tetrahedron whose edge lengths are given by the $\{l_i\}$ and we have substituted the value of the dihedral angle using (21).

Using the fact that the Einstein Hilbert action with negative cosmological constant evaluated on a hyperbolic space $H$ produces

$$S_{EH} = -\frac{1}{16\pi G_N} \int_H d^3x \sqrt{g}(R - 2\Lambda) = \frac{V(H)}{4\pi G_N} = \frac{V(H)}{\pi b^2}, \tag{23}$$

where we have also used the holographic relation of the central charge $c \approx 6/b^2 = 3l_{AdS}/(2G_N)$ [65], and that the AdS radius for the hyperbolic tetrahedron is chosen to be $l_{AdS} = 1$.

The second term in the exponent of (22) can be written as

$$\sum_i \frac{\theta_i l_i}{2\pi b^2} = \sum_i \frac{\theta_i l_i}{8\pi G_N} = \frac{1}{8\pi G_N} \sum_i \int_{\Gamma_i} \theta_i \sqrt{h}, \tag{24}$$

where $\Gamma_i$ denotes the edges of the tetrahedron and $h$ the induced metric on it. This is almost the same as the Hayward corner term introduced along a codimension two surface [66, 67], which corresponds here to each of the edges of the tetrahedron. Let us focus on an edge running orthogonal to the boundary surface. Multiple tetrahedra would share the edge. This edge has to be weighted by the Plancherel measure before integrating over its label. These internal edges orthogonal to the boundary surface are geodesics that run all the way down to the "tip" of the hyperbolic tetrahedron. The quantity in the exponent appears to be related to the "half-space" entanglement entropy by the Ryu-Takayanagi formula and should be related to the Plancherel measure. Indeed, one can check that in the $b \to 0$ limit at finite $l$, the Plancherel measure satisfies [68],

$$\ln \mu(P(l)) \approx l/(4G_N), \qquad P = \frac{l}{2\pi b}. \tag{25}$$

Therefore we recover $\exp(-l_\sigma/(4G_N)) \approx \mu(P_\sigma)^{-1}$.

The factor of the Plancherel measure allocated to each internal edge and the $6j$ symbol together contributes to the factor

$$\exp\left( -S_{EH} + \frac{\sum_e (2\pi - \Theta_e) l_e}{8\pi G_N} + \text{surface terms on } \partial H \right), \tag{26}$$

where $\Theta_e = \sum_i \theta_i$ i.e. the sum over dihedral angles of tetrahedra sharing this internal edge $e$, as shown in figure 6. Thus $2\pi - \Theta_e$ is the angle deficit of the gluing. This term vanishes when the gluing is smooth such that $\Theta_e = 2\pi$. Interestingly it is the shrinkable boundary that was behind this cancellation. There is not any co-dimension 1 term corresponding to the Gibbons-Hawking-York terms for these internal surfaces exactly when the gluing is smooth, as it should

---

[10]In any event, this result would correspond to one of the saddles of the $6j$ symbols in the $b \to 0$ limit [58].

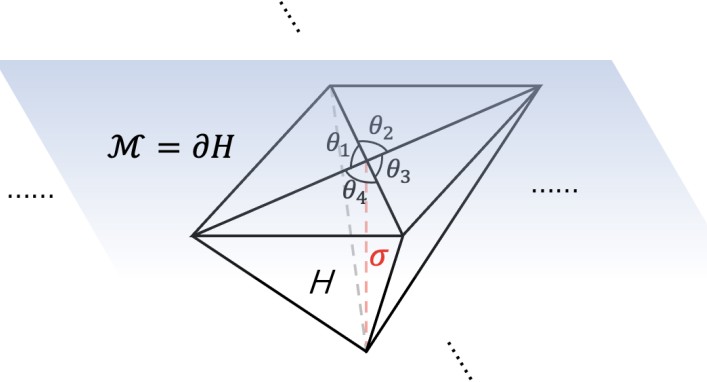

Figure 6: The triangulation that is implied in (16), where each tetrahedron has one surface on the boundary $\mathcal{M} = \partial H$. Three edges of each tetrahedron points in a direction orthogonal to the boundary surface. Several tetrahedra shares one edge, such as the one colored red and labelled by $\sigma$. The conical deficit around this edge is given by $2\pi - \Theta_\sigma$, where $\Theta_\sigma \equiv \theta_1 + \theta_2 + \theta_3 + \theta_4$, and $\theta_i$ are dihedral angles.

be in the gravitational action. Therefore in the classical limit that we are taking, the wavefunctions of (16) reduces to $\exp(-S_{EH}(H))$ where $H$ is the overall 3D hyperbolic space obtained from smoothly gluing all the tetrahedra together, and $\partial H = \mathcal{M}$ by construction. The full Liouville path-integral would require integrating over all $P$'s, which corresponds to summing over tetrahedra of all shapes and sizes. In the small $b$ limit this integral should admit saddle point approximations, which corresponds to solving the Einstein equation globally. While this is beyond the current paper, we will comment on the connection between UV cutoff and IR cutoff surface in the AdS boundary in the saddle point approximation in the conclusion section. We have not said much about the surface terms on $\partial H$ that follows from the boundary condition $\langle\Omega|$, which we hope to return to in the future.

## 4  Renormalisation operator and holographic geodesic tensor network

In the strange correlator construction of the CFT partition function, a specific choice of triangulation is made, which is encoded in the boundary surface of the 3D manifold. As discussed in section 2, this triangulation can be changed via crossing relations, and with the shrinkable boundary, adding/removing bubbles.

One could also choose to consider the re-triangulation of $\langle\Omega|$ and $|\Psi_{\mathcal{U}_q(sl(2,\mathbb{R}))}\rangle$ separately. The pentagon relation and the orthogonality condition satisfied by the $6j$ symbols allow one to relate wave functions $|\Psi_{\mathcal{U}_q(sl(2,\mathbb{R}))}\rangle$ with different *boundary surface triangulations*, as illustrated in figure 3.

Consider a wavefunction defined on a regular lattice with triangle edge lengths given by $\Lambda$, and another on a rescaled lattice with edge lengths $\lambda\Lambda$, these can be mapped to each other by an RG operator $U(\lambda)$, constructed from the $6j$ symbols via recursive use of the relations in figure 3. Part of the RG operator that produces a coarse-graining to the surface wave-function which removes one boundary vertex is illustrated in figure 7. Therefore,

$$|\Psi_{\mathcal{U}_q(sl(2,\mathbb{R}))}\rangle_\Lambda = U(\lambda)|\Psi_{\mathcal{U}_q(sl(2,\mathbb{R}))}\rangle_{\lambda\Lambda}, \tag{27}$$

i.e. The action of $U(\lambda)$ changes the lattice scale by a constant factor. For example in figure 5

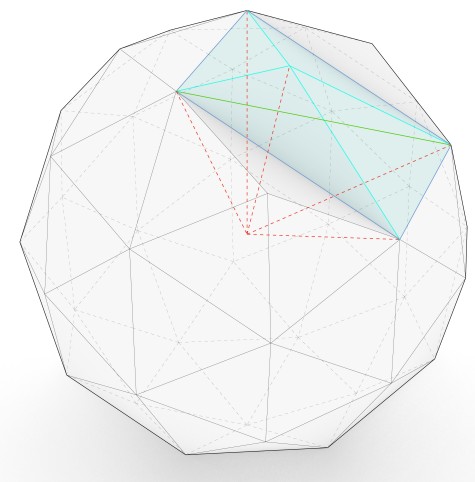

Figure 7: Another 3D view of the wave-function (16). Using the pentagon identity, we can change the triangulation both in the bulk and on the boundary. This is indicated by the shaded pair of tetrahedron which is essentially mapping the original surface links of the sphere (shaded cyan) to a "coarse grained" link shaded green. The shaded pair of tetrahedron is precisely part of the RG operator $U(\lambda)$ in equation (27) in 3D. The RG step we illustrate here matches with what is illustrated in figure 5 (b).

the lattice spacing increases by a factor of $\lambda = \sqrt{2}$ after the bubble is removed, assuming the triangles are isosceles right-angled triangles. This approach has been considered in the context of rational TQFTs [31,69,70], and the corresponding linear map has been studied extensively in [3]. It has been noted that the recursive application of $U(\lambda)$ produces a holographic tensor network that is geometrically a discretisation of a hyperbolic space by tetrahedra. In the present case, $U(\lambda)$ is constructed from $\mathcal{U}_q(sl(2,\mathbb{R}))$ 6$j$ symbols, such that each tetrahedron represents the quantum path integral over a hyperbolic tetrahedron, with its edges being geodesics in 3D hyperbolic space. The strange correlator can thus be written as

$$\langle \Omega(\Lambda)|\Psi_{\mathcal{U}_q(sl(2,\mathbb{R}))}\rangle_\Lambda = \langle \Omega(\Lambda)|U^N(\lambda)|\Psi_{\mathcal{U}_q(sl(2,\mathbb{R}))}\rangle_{\lambda^N \Lambda}. \tag{28}$$

The 2D boundary condition $\langle \Omega(\Lambda)|$ is not topological. Consequently, $\langle \Omega(\Lambda)|U^N(\lambda)$ connects boundary conditions at different lattice scales and helps track the depth in the radial direction. This happens exactly as in the usual AdS/CFT correspondence, where the boundary condition governs the couplings of the QFT. We note that this flow is closely related to the numerical tensor network renormalisation procedure developed in [71–74]. If the boundary is not already at a CFT fixed point, the introduction of dimensionful perturbations causes these couplings to flow under the bulk radial evolution, inducing a symmetry-preserving RG flow. The radial Wheeler-DeWitt equation, which represents the infinitesimal form of (27), thus reproduces the Callan-Symanzik equation of QFTs [75–77]. A similar consideration in the present discrete formulation also reproduces this equation, as further explored in detail in [14].

The Liouville partition function can thus be expressed as an explicit holographic tensor network $U^N(\lambda)$, consisting of a network of geodesics connecting holes at the boundary surface. These geodesics go deeper into the bulk if they connect holes separated by longer distances at the boundary surface, as illustrated in figure 8. Geodesic lengths serve as natural degrees of freedom in this formulation of the bulk.[11]

---

[11] This may be related to the program proposed in [78]. Additionally, coarse-graining of spin networks has been

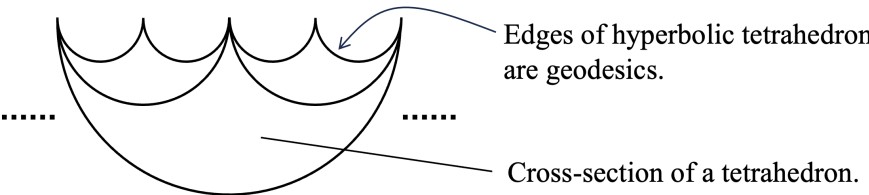

Edges of hyperbolic tetrahedron are geodesics.

Cross-section of a tetrahedron.

Figure 8: The 2D cross-section of $U^N(\lambda)$.

We note that this bears profound resemblance to the tensor network reconstruction of p-adic CFT partition function [80, 81]. There, the boundary CFT is quasi-1d, and the bulk holographic tensors are constructed using the structure coefficients of the p-adic CFT which forms an associative algebra. Here, to reconstruct a CFT that is 2D, the holographic tensor network is constructed out of *open structure coefficients* of the CFT, and the mathematical structure involved is a fusion tensor category, which can be considered as a categorification of an associative algebra.

## 5 Holographic tensor network for wave-functions on a time slice

It has been proposed in the famous work [45] that the CFT wave-function should admit a holographic tensor network description to account for the Ryu-Takayanagi formula [82, 83]. Holographic tensor network describing a state in the CFT is a particularly active area of research.[12] The framework we employ here naturally constructs path-integral of CFTs on a 2D surface. Therefore we can prepare any state by performing a path-integral on manifolds with boundary. For example, to prepare ground state of a CFT state on a circle, one could perform a standard path-integral on a disk. In our framework, we need to pick a triangulation, and one can for example choose to triangulate the disk like cutting a pie into wedges, as shown in figure 9 (a). Notably, [102] proposed the idea of preparing a tensor network using BCFT, which is essentially what is being realised here with specific choice of shrinkable boundary suited for reproducing the quantum Liouville theory. The boundary of the disk would carry dangling legs carrying indices of the open-primaries and their descendants, and also conformal boundary conditions. It has been shown in [4] that the space spanned by these indices can indeed reproduce the closed spectrum, consistent with the fact that the holes do close when the half boundaries are completed into a closed curve.

Recall that open correlation functions are associative as shown in figure 5. Holes can also be added or removed freely using the shrinkable boundary. Therefore in principle we can choose any triangulation of the disk and any triangulation can be converted to any other. In particular, one can convert this canonical triangulation in figure 9 (a) into a tree tensor network by recursive use of the associativity condition, producing an explicit holographic tensor network that guarantees to reproduce the CFT wave-function exactly, as illustrated in figure 9 (b). This is the same idea employed in our construction of the topological RG operator implemented in 2D [3]. The tensors are not unitaries, but they behave like isometries in a perfect tensor code as expected in holographic tensor networks [45, 84]. The key property is again the shrinkable boundary (10), which ensures that two triangles in the shaded region in figure 5 would be reduced to a simple strip. And we note that the pair of triangles can share *any two* edges, and the reduction still applies. This is the analogue of the "perfect" tensor con-

---

explored in the loop quantum gravity literature, such as in [79], which shares notable similarities with the current discussion.

[12]See for example, [45, 78, 84–101].

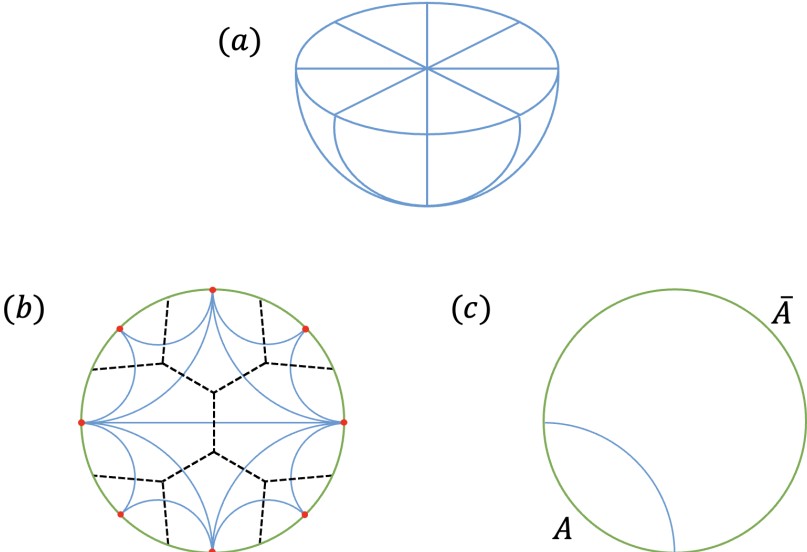

Figure 9: (a) One convenient triangulation of the disk and its interpretation in 3D. Note that these wedges are hyperbolic tetrahedra in 3D. (b) Systematic re-triangulation via use of associativity condition, which converts the wave-function into a holographic tensor network. (c) Pick a triangulation so that one of the links of the holographic network cut the disk separating region $A$ from $\bar{A}$ when computing entanglement entropy.

dition [84], the latter of which requires that for an arbitrary majority set of legs contracted between a perfect tensor and its conjugate, it produces an identity operator. Let us emphasize here however in this exact tensor network that reproduce the CFT rigorously, these tensors are *not* perfect tensors after all. They are very precise three point correlation functions that can be explicitly computed in a given CFT, as has been illustrated in [4]. Instead of the identity operator, the result of the contraction produces the propagator of the state at the RT surface (with boundaries $\sigma_{1,2}$), which is $\exp(-\tau H_{open}^{\sigma_1 \sigma_2})$ for some $\tau$ depending on the details of the size of the region. It would be evident shortly that $H_{open}^{\sigma_1 \sigma_2}$ is indeed the modular Hamiltonian. We call this current reduction a "quasi-perfect-isometric" (QPI) condition. It ensures that the modular Hamiltonian has a non-trivial spectrum.

Note also that here the RT surface does *not* follow from the counting of minimal number of cuts through the graph. This is meaningless since our tensor network is *graph independent*. Instead, the RT surface is measuring the total topological charge following from the fusion product of all the charges at the boundary segment. This is a robust physical quantity that does not depend on how we discretise the bulk. This is the main departure from the perfect tensor network in [84]. In fact, it has been pointed out in recent discussions that the RT formula cannot simply arise from counting cuts in a tensor network, but should instead be related to topological charges [55, 103].

We note that for any specific choice of graphs, it is not possible to access arbitrary local information in the bulk. This reflects the well-known issue of the lack of sub-AdS locality in tensor network representations of the CFT wavefunction [104]. However, in our case, the "tensor network" provides a faithful representation of the wavefunction, and the triangulation remains flexible, allowing it to be adapted to any convenient choice depending on the computation. Therefore we are essentially keeping an infinite number of graphs and we have access to any geodesics connecting two points in the boundary. As it is noted for example in [105, 106],

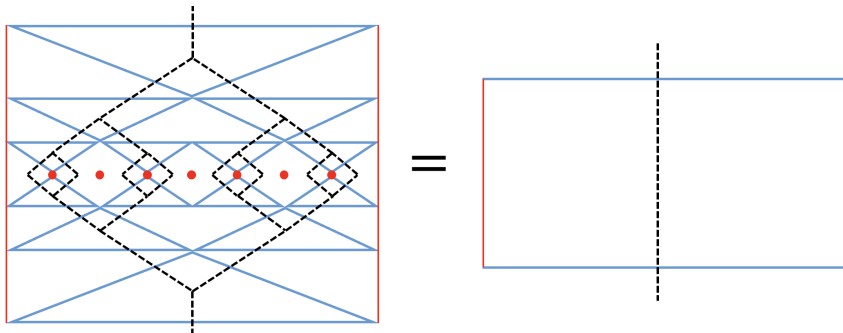

Figure 10: The associativity condition can be used to turn the CFT states prepared by Euclidean path integral into a holographic tensor network. In performing the partial trace for the calculation of Renyi entropy, the pairs of triangles will be reduced to simple strips. This property is similar to isometry of the perfect tensors [84]. Note that each loop in the dual graph actually contains only one vertex – the vertices of the triangles inside the loop are actually all identified, and therefore the contraction only merits one red dot.

these boundary geodesics should provide sufficient data to access any bulk curves,[13] and we believe that is why sub-AdS locality is preserved.[14] The QPI condition in the tensor network representation geometrizes the isometric property of the bulk to boundary map in this case.

Now consider computing entanglement entropy. For simplicity, consider the entanglement entropy of one connected interval $A$. We can always pick a holographic network so that one of the edge cuts through the disk, separating the region from its complement, as shown in figure 9 (c). This edge by construction is a geodesic in the bulk, and thus a Ryu-Takayanagi (RT) surface for this boundary region $A$. Now when computing reduced density matrix $\rho_A$ by tracing out the complement of $A$ between two copies of the wave-function in figure 9 (c), the QPI condition would ensure that the outer layers of triangles contract and reduce to strips that are merged when holes are closed, until we reach a triangle whose bottom edge is the RT surface for $A$, with edge length $P$ labelling the open CFT state propagating through that edge. We note that this picture actually works also for generic RCFTs, which is perhaps not surprising given that the single interval entanglement in 2D CFT is universal. One difference there is that the open state label at the RT surface would not generically adopt an interpretation as the edge length of a hyperbolic geodesic.

When one computes the Renyi-entropy by $S_n(A) = \frac{1}{1-n}\ln(\frac{\text{tr}(\rho_A^n)}{\text{tr}(\rho_A)^n})$, the factor $\text{tr}(\rho_A^n)$ corresponds to a computation that is depicted in figure 11.

It immediately reduces to $\exp(2n\pi i\tau' H_{open}^{\sigma_1\sigma_2})$, where $H_{open}^{\sigma_1\sigma_2}$ is the open state Hamiltonian acting on the open state on the RT surface with boundary conditions $\sigma_{1,2}$, and $\tau'$ depends on the ratio of the size of the hole to the size of $A$. This is in fact a cylinder with the two boundaries to be closed by the shrinkable boundary. In the RCFT case, the shrinkable boundary gives

$$\text{tr}(\rho_A^n) = \langle\langle 0|e^{-2\pi i/(n\tau)H_{closed}}|0\rangle\rangle = \chi_0(-1/(n\tau)), \tag{29}$$

reducing to a standard result. In the open state channel this is equal to

$$\chi_0(-1/(n\tau)) = \sum_i S_{0i}\chi_i(n\tau). \tag{30}$$

---

[13]The two dimensional version of the above observation involving Wilson loops on the boundary and extremal surfaces in the bulk is presented in [107,108].

[14]We thank Tadashi Takayanagi for raising the question of sub-AdS locality in our tensor network and for helpful discussions on this point.

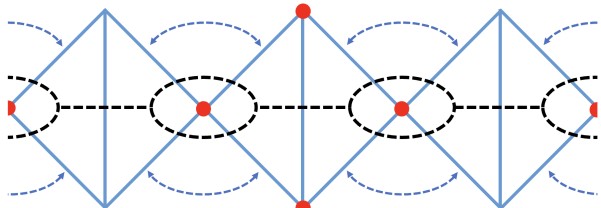

Figure 11: Computation of Renyi entropy for $n = 3$ reduces to contraction of three copies of the reduced density matrix which is depicted by each of the tilted square. Region $A$ is chosen to be made up of two edges here, and the verticle edge corresponds to the RT surface. We place only 1 red dot each at the top and bottom of the picture, signaling that on each side these vertices are identified, and make up only one hole.

The shrinkable boundary thus implies a specific density of states of the open states (which are eigenmodes of the modular Hamiltonian) that is being summed over. The standard result of single interval Renyi entropy and entanglement entropy for RCFTs follows from this result [1, 109]. Note that the calculation for single interval entanglement bears many resemblance to [55], except that here the bulk theory is explicitly united with the CFT, and the edge modes are directly packaged as open CFT states.

Multi-intervals entanglement entropies are not as universal, and they depend on a lot more details of the CFT, which is similar to the analysis for perfect tensor codes [84].

For Liouville theory, single interval entanglement entropy of the ground state is not very well defined, since the vacuum state is not normalisable and thus excluded from the physical spectrum. Extra regularisation is needed such as by introducing subtraction by a reference state [110]. Naively doing the computation in our formulation indeed produces a divergent result.

We will leave a more general study of entanglement entropy using our tensor network, including the way of reproducing the universal multi-interval Renyi entropy predicted by the Ryu-Takayanagi formula in 2D holographic CFTs [111, 112] to a separate publication.

## 6 Conclusions and outlook

In this paper, we have shown that the path-integral of quantum Liouville CFT can be expressed as a discrete state-sum. This is made possible by the shrinkable boundary we proposed here that closes holes in the 2D path-integral. The path-integral written in this form can be written as a strange correlator $\langle \Omega | \Psi_{\mathcal{U}_q(sl(2,\mathbb{R}))} \rangle$, much like the case of RCFTs [3, 4]. This wave-function is a Turaev-Viro-like formulation of an irrational TQFT based on representations of $\mathcal{U}_q(sl(2,\mathbb{R}))$. We do not claim to produce a complete Turaev-Viro formulation of an irrational TQFT because these $6j$ symbols are not expected to allow for a 1-4 move (i.e. adding a bulk vertex in the bulk, and turning 1 tetrahedron into 4),[15] but only the 2-3 move (as guaranteed by the pentagon equation, which is the BCFT crossing relation) and removal of pairs of tetrahedra by the orthogonality condition. Without the 1-4 move we cannot guarantee to triangulate the 3-manifold in completely arbitrary ways. However, keeping only the pentagon relation and orthogonality was enough to reproduce the CFT path-integral while allowing the CFT path-integral to be triangulated in arbitrary ways.[16] We produce a canonical triangulation of the 3D bulk that is well behaved locally and seems to be so globally whenever the Liouville path-

---

[15]We thank Mengyang Zhang for discussions on this issue. We note that the 1-4 move is the cause of divergence suffered by other constructions of irrational TV state sum as discussed in [13].

[16]We can add boundary vertices and get a convergent answer, but not the bulk vertices.

integral is itself expected to be so too. We expect that this wave-function should be related to the Teichmuller TQFT [43] and the recently proposed Virasoro TQFT [13, 59]. And interestingly, in the large central charge limit, using the asymptotics of the quantum $6j$ symbol for certain representations of $\mathcal{U}_q(sl(2,\mathbb{R}))$, we show that the wavefunctions of $|\Psi_{\mathcal{U}_q(sl(2,\mathbb{R}))}\rangle$ reduces to $\exp(-S_{EH})$, where the Einstein Hilbert action is evaluated on a hyperbolic space $H$ which came from gluing hyperbolic tetrahedra together. The result also extends beyond the large $c$ limit, enabling a non-perturbative summation over quantum geometries. By construction, the boundary of $H$ is the 2D surface on which the CFT path-integral is performed. The full Liouville path-integral instructs the precise sum over the constituent tetrahedra of $H$ of different shapes and sizes that would faithfully reproduce itself. This construction also naturally produces a geodesic network which is the set of variables of the 3D bulk theory.

There are several important applications and many interesting future problems that we would like to discuss to different level of details here.

### 6.0.1 Boundary conditions, the UV/IR cutoff and RG

We have essentially given a specific boundary condition $\langle\Omega|$ to a path-integral on a 3D manifold with boundary, $|\Psi_{\mathcal{U}_q(sl(2,\mathbb{R}))}\rangle$. We have shown that $|\Psi_{\mathcal{U}_q(sl(2,\mathbb{R}))}\rangle$ reduces to the exponentiation of the Einstein Hilbert action in the bulk in the large $c$ limit, but we have not made precise comparisons of the boundary conditions at the boundary of the manifold with the AdS/CFT prescription. We note that the boundary condition alters the bulk by changing the weight of contributing geometries as they are summed. For example, it is clear that the precise saddle point of the path-integral depends sensitively on the boundary condition. It is possible to show that the size of the hole is inversely related to the saddle point of the geodesic length connected to the hole.

Let us illustrate with a simple example. Consider the bubble with two legs considered in (10). While we computed the exact result there, we could ask for the saddle point in the small $b$ limit for $P_k$. The integral of $P_k$ was to be completed before the last equality. The $k$ bubble conformal block could be approximated by $\chi_{P_k}(\tau)$ in the small hole ($\tau = i\epsilon$ and $\epsilon \to 0$) limit since the decendants from the identity line connected to the bubble should be suppressed. We have

$$\chi_{P_k}(\tau) = \frac{e^{2\pi i \tau P_k^2}}{\eta(\tau)}. \tag{31}$$

We can compute the saddle point for $P_k$ in the integral $\int_0^\infty dP_k \mu(P_k) \frac{e^{2\pi i \tau P_k^2}}{\eta(\tau)}$ in the limit $b \to 0$. The saddle point, to leading order in small $b$, is located at

$$P_k^* = \frac{1}{2b\epsilon}. \tag{32}$$

We note that the hole is closed up by the "vacuum" Ishibashi state. The descendant states in the Ishibashi state are suppressed by $\exp(-\frac{2\pi}{\epsilon}\hat{D})$. When $\epsilon$ is small but finite, the first descendant state $L_{-2}\bar{L}_{-2}|0\rangle$ would contribute non-negligibly. The length of the geodesic is cut-off essentially by the introduction of $T\bar{T}$ deformation [2, 113], which matches with the expectation from the AdS/CFT correspondence [77]. As mentioned in section 4, $\langle\Omega|$ controls couplings of the 2D theory. The introduction of finite sized holes is one possible and convenient way of introducing $T\bar{T}$ deformation, which has a particularly simple integrable flow under RG that admits a holographic interpretation of radial evolutions [77]. This flow is precisely driven by the RG operator $U(\lambda)$ introduced in section 4, and has since been discussed in detail in [14].

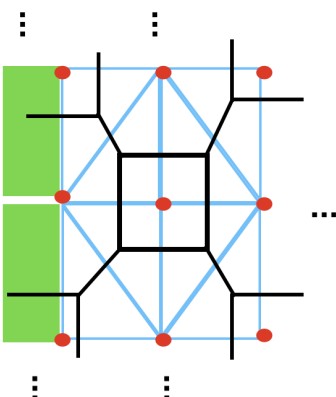

Figure 12: Boundary tensors representating a conformal boundary conditions are represented by these green blocks. Generally they are direct product states carrying indices of the vertex variables and the open primaries and descendants.

### 6.0.2 CFT partitions on surfaces of higher genus and operator insertions

Our construction of the 2D path-integral involves tiling a flat surface with flat triangles for the given conformal map we referred to and detailed in [4]. To explicitly work with (compact) surfaces of more general genus while sticking to flat tiles, one could make use of quotients by the classical Schottky groups that was considered when constructing appropriate AdS bulk whose boundary is the desired higher genus surface [114]. Here, we are inverse engineering from the boundary.

Another possibility is to change the boundary condition and instead of mapping the three point function to flat triangles, one could also consider mapping it to hyperbolic frame, and obtain compact Riemann surfaces with punctures via orbifolding by the Fuchsian group. Such different conformal frames chosen in the boundary conditions $\langle\Omega|$ should be the analogue of choosing appropriate induced metric on the cut-off surface in AdS/CFT. To control the precise operator insertion, the simplest method is to insert operators in one of the holes using the bulk-boundary two-point functions at the vertex of the surface triangulation. Instead of using the Plancherel measure as the measure in the integral over boundary conditions, we can replace the measure by $S_{ij} = 2\sqrt{2}\cos(4\pi P_i P_j)$ i.e. other components of the modular matrix instead of the $S_{\mathbf{1}i}$ component. This allows one to insert more general primaries at the holes. It is interesting that the prescription for operator insertion appeared also in [115] when the authors defined the overlap of an RCFT correlation with a TV state.

### 6.0.3 AdS/BCFT

The building blocks of the CFT path-integral are three point open correlation functions. It is evident that the corresponding dual geometry is a tetrahedron which is a hyperbolic space with boundary. The AdS/BCFT dictionary proposes that conformal boundary condition would lead to a brane that extends into the AdS bulk [116]. This expectation is indeed played out in our construction. One trivial way of producing CFT path-integrals with boundaries of different sizes and shapes is to pick a conformal block in an appropriate conformal frame in $\langle\Omega|$.

There is another way of producing conformal boundary conditions. In the tensor network literature, it is known that conformal boundary conditions correspond to special set of "fixed point boundary tensors" as illustrated by the green blocks in figure 12, that can be searched numerically [117–119]. Here, it is also possible to construct these fixed point boundary tensors from correlation functions of the CFT. They allow one to produce individual building block to

tile a boundary of arbitrary sizes and shape without changing the choice of conformal frame of the boundary conditions for every new situation. This should allow one to study saddle points of the boundary extending into the bulk in a setting closer in spirit to AdS/BCFT boundary brane proposal. We will report on these results in a separate publication.

### 6.0.4 Other irrational CFTs and ensemble average

The current construction depends on complete knowledge of the conformal boundary conditions of the CFT. For RCFTs it is well known that knowledge of the boundary conditions correspond to selecting a modular invariant CFT [5,120]. Here, the boundary conditions essentially select the choice of geometries to be included in the path-integral. In this case, the CFT does not involve an ensemble average.

This back-engineering employed for Liouville theory is not easy to replicate in general irrational CFTs because there is very limited examples where the full quantum CFT including their chiral symmetry preserving conformal boundary conditions are known completely. While the Liouville theory is generically the *effective* theory for irrational CFTs, the spectra of Cardy boundary conditions would still depend on the original theory, and thus specify a sum over geometries that is different from the current example. To proceed without full knowledge of the boundary conditions, one needs to make an ansatz of the spectra of boundary conditions. For example, in a generic holographic CFT expected to have a big gap, the Cardy state as a sum over Ishibashi states constructed from differemt primaries would also need to reflect that [121].

In addition to specific modular invariant CFTs, one can also consider ensemble averaged CFTs. Using the random tensor ansatz that generalizes the eigenstate thermalisation hypothesis (ETH) and introducing ensemble averaging [122–126], we can ensemble average the open structure coefficients that enters the CFT partition function [121,127,128]. This is analogous to the considerations in [125,126] that are applied to closed correlation functions.[17] In the current framework, ensemble averages of the open structure coefficient generate an alternative random tensor network, which produces a different sum over geometries, suggesting a universal emergence of AdS geometries. We want to emphasize that the ensemble average is also related to Liouville theory, but as an effective field theory for the stress tensor sector of holographic CFTs. This is different from a bona fide UV complete quantum field theory on its own as considered in this paper. We will report about these results elsewhere.

### 6.0.5 Lorentzian signature and black holes

It is one important purpose of the current construction to extract new insights on black hole physics. In particular it would be very interesting to explore physics behind the event horizon from the perspective of the CFT. To that end, it is necessary to continue the path-integral to Lorentzian signature.

Naively, Liouville theory may not seem well-suited for studying black hole physics. Its flat spectrum and the absence of a vacuum state in the physical spectrum prevent it from directly describing black hole microstates [49–52]. However, there are compelling indications that Liouville theory can still provide valuable insights into black hole physics.

First, as observed in [130], the Hartle-Hawking state for eternal black holes in 3D can be described using Liouville theory by employing two copies of the ZZ boundary states [131], extending earlier proposals that used boundary states to represent thermal states [132,133]. Additionally, scattering processes in AdS$_3$ black holes and out-of-time-ordered correlation functions are universally captured by the $\mathcal{U}_q(sl(2,\mathbb{R}))$ $6j$ symbols utilized in this paper [51,134].

---

[17]We were informed that [126] may also lead to a similar simplicial representation of the bulk theory based on quantum $6j$ symbols [129].

In fact, even without studying the dynamics in Lorentzian signature, we can already learn something about black hole physics by looking at the $t = 0$ time slice, using states dual to black holes prepared by CFT path integral [135–139]. In particular, we can find exactly the position of the horizons at $t = 0$ in gravity by studying the CFT path integral, and use the tensor network to go into the bulk understand the properties of the horizon. The exact tensor network might help us understand how the map becomes non-isometric in the semi-classical limit [140, 141]. We can also probe sub-AdS physics for the black holes by adding dressed bulk probe particles using [142–144], and even model the effect of Hawking radiation by inserting a pair of operators [137]. We will report more results in another publication.

### 6.0.6  Other RCFTs and AdS/CFT

The current construction of CFT path-integrals were initially introduced for rational CFTs. Quantum $6j$ symbols for representations of each compact quantum group can be used to construct different RCFTs. We note it is not only quantum $6j$ symbols $\mathcal{U}_q(sl(2,\mathbb{R}))$ associated to Liouville theory discussed in this paper that admits connection to geometrical volumes. It is known that quantum $6j$ symbols associated to $\mathcal{U}_q(sl_2)$ where $q = \exp(2\pi i/N)$ admits a large $N$ limit, which are also connected to volumes of tetrahedra [145, 146]. It is perhaps possible to connect more generic RCFTs to a sum over geometries. It is also plausible that all holographic CFTs contain a universal sector that is related to sum over geometries.

### 6.0.7  Connection with lattice models

Our framework was inspired by the strange correlator construction of integrable lattice models in [31, 147]. As it is observed in [3], the boundary conditions $\langle\Omega|$ chosen in [31, 147], which are direct product states, can be understood as seed states which would flow to a fixed point under repeated use of the RG operator [3] reviewed in section 4 above, when the seed state is fine-tuned to a critical coupling. The fixed point of the flow is precisely constructed from 3-point conformal blocks in [3, 4].

It is known that there is an integrable lattice Liouville theory [148, 149][18] which shares many similar structures as the infinite family of lattice integrable models discussed in [31, 147]. We believe the transfer matrix of the lattice Liouville model could also be written as a strange correlator in a similar manner, and they would provide a direct product seed state similar to the case of integrable lattice models. The end point of the RG flow when the seed state is at a critical point should be the bra state $\langle\Omega|$ in this paper.

### 6.0.8  Higher dimensions

The strange correlator construction of lattice models can be generalised to arbitrary dimensions. For example, it is observed that partition functions of lattice models such as the 3D Ising model can also be expressed as a strange correlator $\langle\Omega|\Psi\rangle$, where $|\Psi\rangle$ corresponds to the ground state of a 4D TQFT [3]. In these lattice models, $\langle\Omega|$ is a direct product state. At the critical point $\langle\Omega|$ should also flow into an eigenstate constructed from the RG operator that follows from the 4D TQFT [3]. However, so far we do not have any analytic understanding of the fixed point of the RG operator like what we have for 2D CFTs. We believe analogous to the 2D case the eigenstates of 4D RG operators should also be constructed from conformal blocks involving boundaries and condimension 2 defects. If we have a better understanding of these objects it would be possible to replicate the success of 2D CFTs in 3D, and perhaps also in CFTs in even higher dimensions. Whether this is still connected to the AdS/CFT is unclear – although given the similarity in structure between the 2D and 3D strange correlators at the

---

[18]We thank Herman Verlinde for pointing us to this model whose connection with chaos is discussed in [150].

level of the lattice models, it is not impossible that the higher dimensional strange correlators also carries connection with the AdS/CFT correspondence.

To end our discussions, we acknowledge that Liouville CFT is a perculiar theory that does not contain a vacuum state, nor states carrying non-trivial spins, and has a flat density of states. Its saddles would miss out on some of the most interesting geometries in gravity. Whether or not we call the holographic dual of Liouville theory "quantum gravity" that people believe should carry certain properties, it is indeed a precise UV complete quantum path-integral over geometries. It serves as a proof of principle that a UV-complete path integral can be constructed to recover a specific CFT exactly, rather than an ensemble average or coarse-graining of CFTs. It also suggests that perhaps a UV complete quantum measure of quantum gravity is not unique. Whether or not it carries various properties we expect—such as including favored geometries like the BTZ black hole as dominant saddles—likely depends on the specific CFT dual being reconstructed. This scenario is already realized in the case of holographic duals of RCFTs, where the same topological bulk can correspond to different modular invariants of the CFT, achieved by altering the quantum measure controlled by a *Lagrangian algebra* of topological lines [2, 4, 5, 12]. At present, there are no known obstructions to constructing holographic duals, potentially incorporating gravity and other matter, for other irrational CFTs using our strategy of discretization and the search for shrinkable boundaries. Our framework holds the potential for broader applicability, offering a microscopic translation between CFT and bulk degrees of freedom, while also establishing an explicit and computable connection to tensor networks, as demonstrated in the case of Liouville theory. We hope that such constructions will provide fresh insights into the AdS/CFT correspondence and black hole physics in the near future.

## Acknowledgments

We thank Ning Bao, Scott Collier, Muxin Han, Sarah Harrison, Henry Lin, Nicolai Reshetikhin, Martin Sasieta, Sahand Seifnashri, Yidun Wan and Hao Zheng for helpful discussions. We thank Bin Chen, Wan Zhen Chua, Henry Maxfield, Thomas Mertens, Tadashi Takayanagi, Herman Verlinde, Zixia Wei, Gabriel Wong and Mengyang Zhang for comments and discussions on the draft. We thank Gong Cheng and Zhengcheng Gu for collaborations on related projects. We thank Zhenhao Zhou for plotting the 3D diagram in this paper. YJ thanks his friends for encouraging him to continue pursuing research in physics.

**Funding information** LYH acknowledges the support of NSFC (Grant No. 11922502, 11875111). LC acknowledges support from the NSFC Grant No. 12305080, the Guangzhou Science and Technology Project with Grant No. SL2023A04J00576, and the startup funding of South China University of Technology. YJ acknowledges the support by the U.S Department of Energy ASCR EXPRESS grant, Novel Quantum Algorithms from Fast Classical Transforms, and Northeastern University. BL thanks the Laboratoire de Physique Théorique et Hautes Energies for the hospitality.

# A  Normalization and structure coefficients of bulk and boundary operators in Liouville theory

## A.1  Two and three point functions of bulk Liouville operators

We will explain the normalization of bulk and boundary Liouville operators used in this paper here.

The usual definition of bulk Liouville primary operators [7] has

$$V_\alpha(z,\bar{z}) = e^{2\alpha\Phi(z,\bar{z})}, \tag{A.1}$$

with conformal dimension $h = \bar{h} = \alpha(Q-\alpha)$, and $\alpha$ is related to the Liouville momentum $P_\alpha$ by $\alpha = \frac{Q}{2} + iP_\alpha$, where $P \in \mathbb{R}$ for normalizable operators [151].

The two point functions of these operators have,

$$\langle V_{\alpha_1}(0)V_{\alpha_2}(1)\rangle = 2\pi[\delta(P_1 + P_2) + S_L(\alpha_1)\delta(P_1 - P_2)], \tag{A.2}$$

where $S_L(\alpha)$ is the Liouville reflection coefficient and is equal to

$$\begin{aligned}
S_L(\alpha) &= -\left(\pi\mu\gamma(b^2)\right)^{-2iP_\alpha/b}\frac{\Gamma\left(1+\frac{2iP_\alpha}{b}\right)\Gamma(1+2iP_\alpha b)}{\Gamma\left(1-\frac{2iP_\alpha}{b}\right)\Gamma(1-2iP_\alpha b)}\\
&= \left(\pi\mu\gamma(b^2)b^{2-2b^2}\right)^{-2iP_\alpha/b}\frac{\Gamma_b(2iP_\alpha)\Gamma_b(Q-2iP_\alpha)}{\Gamma_b(Q+2iP_\alpha)\Gamma_b(-2iP_\alpha)},
\end{aligned} \tag{A.3}$$

here $\mu$ is the cosmological constant in Liouville theory, and $b$ is related to the parameter $Q$ by $Q = b + b^{-1}$.

Following [125, 130], we will define Liouville primary operators using normalization,

$$V_\alpha(z,\bar{z}) \to \frac{V_\alpha(z,\bar{z})}{\sqrt{S_L(\alpha)}}. \tag{A.4}$$

Using this normalization, the normalizable operators in Liouville theory becomes Hermitian operators, since we can identify operators with $P \to -P$, and we choose to focus on $P \geq 0$.

The two point functions using this new normalization becomes,

$$\langle V_{\alpha_1}(0)V_{\alpha_2}(1)\rangle = 2\pi\delta(P_1 - P_2). \tag{A.5}$$

The three point functions in Liouville theory using normalization (A.1) is given by the DOZZ formula [6, 7].

$$\begin{aligned}
C_{\text{DOZZ}}(\alpha_1,\alpha_2,\alpha_3) =&\ (\pi\mu\gamma(b^2)b^{2-2b^2})^{\frac{Q-\alpha_1-\alpha_2-\alpha_3}{b}}\\
&\times \frac{\gamma_0\gamma_b(2\alpha_1)\gamma_b(2\alpha_2)\gamma_b(2\alpha_3)}{\gamma_b(\alpha_1+\alpha_2+\alpha_3-Q)\gamma_b(\alpha_1+\alpha_3-\alpha_2)\gamma_b(\alpha_1+\alpha_2-\alpha_3)\gamma_b(\alpha_2+\alpha_3-\alpha_1)},
\end{aligned} \tag{A.6}$$

where $\gamma(x) = \Gamma(x)/\Gamma(1-x)$. We use the Barnes double gamma function $\Gamma_b(x)$ to define the special functions $\gamma_b(x) = \frac{1}{\Gamma_b(x)\Gamma_b(Q-x)}$ and $S_b(x) = \frac{\Gamma_b(x)}{\Gamma_b(Q-x)}$. The constant $\gamma_0$ is given by $\gamma_0 = \frac{d\gamma_b(x)}{dx}|_{x=0}$. The normalization (A.2) can be gotten from the DOZZ formula with $\alpha$ analytical continued to 0 for one of these operators.

Using our normalization, the three point functions become

$$C(\alpha_1,\alpha_2,\alpha_3) = \frac{C_{\text{DOZZ}}(\alpha_1,\alpha_2,\alpha_3)}{\sqrt{S_L(\alpha_1)S_L(\alpha_2)S_L(\alpha_3)}}. \tag{A.7}$$

This can also be rewritten using the universal OPE coefficient $C_0(P_1, P_2, P_3)$ as [57, 125, 152]

$$C(\alpha_1, \alpha_2, \alpha_3) = \frac{1}{c_b} \sqrt{\rho_0(P_1)\rho_0(P_2)\rho_0(P_3)} C_0(\alpha_1, \alpha_2, \alpha_3), \tag{A.8}$$

where $c_b = \frac{\left(\pi \mu \gamma(b^2) b^{2-2b^2}\right)^{\frac{Q}{2b}}}{2^{\frac{3}{4}}\pi} \frac{\Gamma_b(2Q)}{\Gamma_b(Q)}$, and

$$C_0(\alpha_1, \alpha_2, \alpha_3) = \frac{\Gamma_b(2Q)}{\sqrt{2}\Gamma_b(Q)^3} \frac{\prod_{\pm 1,2,3} \Gamma_b\left(\frac{Q}{2} \pm_1 iP_1 \pm_2 iP_2 \pm_3 iP_3\right)}{\prod_{k=1}^{3} \Gamma_b(Q + 2iP_k)\Gamma_b(Q - 2iP_k)},$$

$$S_{1\alpha} = \rho_0(P_\alpha) = \mu(P_\alpha) = \sqrt{2}|S_b(2\alpha)|^2 = 4\sqrt{2}\sinh(2\pi P_\alpha b)\sinh\left(\frac{2\pi P_\alpha}{b}\right), \tag{A.9}$$

where $\prod_{\pm 1,2,3}$ represents all possible sign permutations in the product of the eight terms.

## A.2 An identity relating modular kernels and $6j$ symbols

For boundary Liouville operators, we will apply the same idea to identify $P \to -P$ and get rid of the boundary reflection coefficients.

Before doing this we will first show an identity relating the modular kernels and $6j$ symbols. We have from [58],

$$
\begin{aligned}
F_{\sigma_2, \beta_3}\begin{pmatrix} \beta_2 & \beta_1 \\ \sigma_3 & \sigma_1 \end{pmatrix} &= |S_b(2\beta_3)|^2 \frac{N(\sigma_2, \beta_1, \sigma_1)N(\sigma_3, \beta_2, \sigma_2)}{N(\beta_3, \beta_2, \beta_1)N(\sigma_3, \beta_3, \sigma_1)} \begin{Bmatrix} \sigma_1 & \beta_1 & \sigma_2 \\ \beta_2 & Q-\sigma_3 & \beta_3 \end{Bmatrix}_b^{an} \\
&= |S_b(2\beta_3)|^2 \frac{N(\sigma_2, \beta_1, \sigma_1)N(\sigma_3, \beta_2, \sigma_2)}{N(\beta_3, \beta_2, \beta_1)N(\sigma_3, \beta_3, \sigma_1)} \frac{M(\beta_3, \beta_2, \beta_1)M(\sigma_3, \beta_3, \sigma_1)}{M(\sigma_2, \beta_1, \sigma_1)M(\sigma_3, \beta_2, \sigma_2)} \\
&\quad \times \begin{Bmatrix} \sigma_1 & \beta_1 & \sigma_2 \\ \beta_2 & Q-\sigma_3 & \beta_3 \end{Bmatrix}_b \\
&= |S_b(2\beta_3)|^2 \frac{N'(\sigma_2, \beta_1, \sigma_1)N'(\sigma_3, \beta_2, \sigma_2)}{N'(\beta_3, \beta_2, \beta_1)N'(\sigma_3, \beta_3, \sigma_1)} \begin{Bmatrix} \sigma_1 & \beta_1 & \sigma_2 \\ \beta_2 & \sigma_3 & \beta_3 \end{Bmatrix}_b \\
&= |S_b(2\beta_3)|^2 \frac{N'(\sigma_2, \beta_1, \sigma_1)N'(\sigma_3, \beta_2, \sigma_2)}{N'(\beta_3, \beta_2, \beta_1)N'(\sigma_3, \beta_3, \sigma_1)} \begin{Bmatrix} \beta_2 & \beta_1 & \beta_3 \\ \sigma_1 & \sigma_3 & \sigma_2 \end{Bmatrix}_b.
\end{aligned}
\tag{A.10}
$$

$\begin{Bmatrix} \beta_2 & \beta_1 & \beta_3 \\ \sigma_1 & \sigma_3 & \sigma_2 \end{Bmatrix}_b$ is the $6j$ symbol for the modular double with full tetrahedron symmetry [58],

$$\begin{Bmatrix} \beta_2 & \beta_1 & \beta_3 \\ \sigma_1 & \sigma_3 & \sigma_2 \end{Bmatrix}_b = \begin{Bmatrix} \beta_1 & \beta_2 & \beta_3 \\ \sigma_3 & \sigma_1 & \sigma_2 \end{Bmatrix}_b = \begin{Bmatrix} \beta_2 & \beta_3 & \beta_1 \\ \sigma_1 & \sigma_2 & \sigma_3 \end{Bmatrix}_b = \begin{Bmatrix} \sigma_1 & \sigma_3 & \beta_3 \\ \beta_2 & \beta_1 & \sigma_2 \end{Bmatrix}_b, \tag{A.11}$$

which can be easily verified by the following explicit expression

$$
\begin{aligned}
\begin{Bmatrix} \alpha_1 & \alpha_2 & \alpha_3 \\ \alpha_4 & \alpha_5 & \alpha_6 \end{Bmatrix}_b &= \Delta(\alpha_3, \alpha_2, \alpha_1)\Delta(\alpha_5, \alpha_4, \alpha_3)\Delta(\alpha_6, \alpha_4, \alpha_2)\Delta(\alpha_6, \alpha_5, \alpha_1) \\
&\quad \times \int_C du\, S_b(u - \alpha_{321})S_b(u - \alpha_{543})S_b(u - \alpha_{642}) \\
&\quad \times S_b(u - \alpha_{651})S_b(\alpha_{4321} - u)S_b(\alpha_{6431} - u)S_b(\alpha_{6532} - u)S_b(2Q - u),
\end{aligned}
\tag{A.12}
$$

with $\alpha_{ijk} = \alpha_i + \alpha_j + \alpha_k, \alpha_{ijkl} = \alpha_i + \alpha_j + \alpha_k + \alpha_l$, and

$$\Delta(\alpha_3, \alpha_2, \alpha_1) = \left(\frac{S_b(\alpha_1 + \alpha_2 + \alpha_3 - Q)}{S_b(\alpha_1 + \alpha_2 - \alpha_3)S_b(\alpha_1 + \alpha_s - \alpha_3)S_b(\alpha_2 + \alpha_3 - \alpha_1)}\right)^{\frac{1}{2}}. \tag{A.13}$$

In addition, the $6j$ symbols satisfy the pentagon identity

$$\int_0^\infty dP_{\delta_1} |S_b(2\delta_1)|^2 \begin{Bmatrix} \alpha_1 & \alpha_2 & \beta_1 \\ \alpha_3 & \beta_2 & \delta_1 \end{Bmatrix}_b \begin{Bmatrix} \alpha_1 & \delta_1 & \beta_2 \\ \alpha_4 & \alpha_5 & \gamma_2 \end{Bmatrix}_b \begin{Bmatrix} \alpha_2 & \alpha_3 & \delta_1 \\ \alpha_4 & \gamma_2 & \gamma_1 \end{Bmatrix}_b = \begin{Bmatrix} \beta_1 & \alpha_3 & \beta_2 \\ \alpha_4 & \alpha_5 & \gamma_1 \end{Bmatrix}_b \begin{Bmatrix} \alpha_1 & \alpha_2 & \beta_1 \\ \gamma_1 & \alpha_5 & \gamma_2 \end{Bmatrix}_b .$$

(A.14)

The normalization factors are given by,

$$N(\alpha_3, \alpha_2, \alpha_1) = \frac{\Gamma_b(2Q - 2\alpha_3)\Gamma_b(2\alpha_2)\Gamma_b(2\alpha_1)}{\Gamma_b(2Q - \alpha_1 - \alpha_2 - \alpha_3)\Gamma_b(Q - \alpha_1 - \alpha_2 + \alpha_3)\Gamma_b(\alpha_1 + \alpha_3 - \alpha_2)\Gamma_b(\alpha_2 + \alpha_3 - \alpha_1)},$$

(A.15)

and

$$M(\alpha_3, \alpha_2, \alpha_1) = \big( S_b(2Q - \alpha_1 - \alpha_2 - \alpha_3) S_b(Q - \alpha_1 - \alpha_2 + \alpha_3) S_b(\alpha_1 + \alpha_3 - \alpha_2)$$
$$\times S_b(\alpha_2 + \alpha_3 - \alpha_1) \big)^{-\frac{1}{2}} .$$

(A.16)

Plugging the formulas we have listed above and after doing some algebra, we get

$$F_{\sigma_2, \beta_3} \begin{pmatrix} \beta_2 & \beta_1 \\ \sigma_3 & \sigma_1 \end{pmatrix} = \sqrt{|S_b(2\beta_3)|^2 |S_b(2\sigma_2)|^2} \sqrt{\frac{C(\beta_3, \beta_2, \beta_1) C(\sigma_3, \beta_3, \sigma_1)}{C(\sigma_2, \beta_1, \sigma_1) C(\sigma_3, \beta_2, \sigma_2)}} \begin{Bmatrix} \beta_2 & \beta_1 & \beta_3 \\ \sigma_1 & \sigma_3 & \sigma_2 \end{Bmatrix}_b ,$$

(A.17)

which takes exactly the same form as in rational CFTs [153]. Notice that both sides of the equation are invariant under Liouville reflection $P \to -P$.

## A.3 Two and three point functions of boundary Liouville operators

Now we will explain our normalization for the boundary Liouville operators, and show that using this normalization, there is a striking formula relating the boundary three point functions and the $6j$ symbols as in rational CFTs.

The usual definition of Liouville boundary operators $\Phi_\alpha^{\sigma_1 \sigma_2}$ have normalization [11],

$$\langle \Phi_{\alpha_1}^{\sigma_1 \sigma_2}(0) \Phi_{\alpha_2}^{\sigma_2 \sigma_1}(1) \rangle = \delta(P_1 + P_2) + S_L(\alpha_1, \sigma_1, \sigma_2)\delta(P_1 - P_2),$$

(A.18)

where $S_L(\alpha, \sigma_1, \sigma_2)$ is the boundary Liouville reflection coefficient,

$$S_L(\alpha, \sigma_1, \sigma_2) = (\pi\mu\gamma(b^2)b^{2-2b^2})^{\frac{1}{2b}(Q-2\beta)} \frac{\Gamma_b(2\alpha - Q)}{\Gamma_b(Q - 2\alpha)} \frac{S_b(\sigma_1 + \sigma_2 - \alpha)S_b(2Q - \alpha - \sigma_1 - \sigma_2)}{S_b(\alpha + \sigma_2 - \sigma_1)S_b(\alpha + \sigma_1 - \sigma_2)} .$$

(A.19)

We will again normalize our operators using

$$\Phi_\alpha^{\sigma_1 \sigma_2}(x) \to \frac{\Phi_\alpha^{\sigma_1 \sigma_2}(x)}{\sqrt{S_L(\alpha, \sigma_1, \sigma_2)}} ,$$

(A.20)

and we will identify $P \to -P$, focusing on $P \geq 0$, with

$$\langle \Phi_{\alpha_1}^{\sigma_1 \sigma_2}(0) \Phi_{\alpha_2}^{\sigma_2 \sigma_1}(1) \rangle = \delta(P_1 - P_2).$$

(A.21)

The three point function with the usual normalization is defined as,[19]

$$\langle \Phi_{\beta_3}^{\sigma_1 \sigma_3}(x_3) \Phi_{\beta_2}^{\sigma_3 \sigma_2}(x_2) \Phi_{\beta_1}^{\sigma_2 \sigma_1}(x_3) \rangle = \frac{C_{\beta_3, \beta_2, \beta_1}^{\sigma_2, \sigma_1, \sigma_3}}{|x_{21}|^{\Delta_1 + \Delta_2 - \Delta_3} |x_{32}|^{\Delta_2 + \Delta_3 - \Delta_1} |x_{31}|^{\Delta_3 + \Delta_1 - \Delta_2}} ,$$

(A.22)

---

[19]Our convention for the three point functions is slightly different and we put labels on the opposite side of an edge on top of the boundary labels, as shown in Figure 2.

and it's given in [11] by Ponsot and Teschner as

$$C_{\text{PT};Q-\beta_3,\beta_2,\beta_1}^{\sigma_2,\sigma_1,\sigma_3} = \frac{g_{\beta_3}^{\sigma_3\sigma_1}}{g_{\beta_2}^{\sigma_3\sigma_2}g_{\beta_1}^{\sigma_2\sigma_1}} F_{\sigma_2,\beta_3}\begin{pmatrix} \beta_2 & \beta_1 \\ \sigma_3 & \sigma_1 \end{pmatrix}, \tag{A.23}$$

where

$$\begin{aligned} g_{\beta}^{\sigma_3\sigma_1} =& \frac{\left(\pi\mu\gamma(b^2)b^{2-2b^2}\right)^{\beta/2b}}{\Gamma_b(2Q-\beta-\sigma_1-\sigma_3)} \\ & \times \frac{\Gamma_b(Q)\Gamma_b(Q-2\beta)\Gamma_b(2\sigma_1)\Gamma_b(2Q-2\sigma_3)}{\Gamma_b(\sigma_1+\sigma_3-\beta)\Gamma_b(Q-\beta+\sigma_1-\sigma_3)\Gamma_b(Q-\beta+\sigma_3-\sigma_1)}. \end{aligned} \tag{A.24}$$

Combining with (A.7) and (A.17) and doing some algebra, we get

$$\begin{aligned} C_{\text{PT};Q-\beta_3,\beta_2,\beta_1}^{\sigma_2,\sigma_1,\sigma_3} =& \frac{1}{\sqrt{\gamma_0}\Gamma_b(Q)} \left(|S_b(2\beta_1)|^2|S_b(2\beta_2)|^2|S_b(2\beta_3)|^2\right)^{1/4} \\ & \times \sqrt{\frac{S_L(\beta_2,\sigma_3,\sigma_2)S_L(\beta_1,\sigma_2,\sigma_1)}{S_L(\beta_3,\sigma_3,\sigma_1)}}\sqrt{C(\beta_3,\beta_2,\beta_1)}\begin{Bmatrix} \beta_2 & \beta_1 & \beta_3 \\ \sigma_1 & \sigma_3 & \sigma_2 \end{Bmatrix}_b. \end{aligned} \tag{A.25}$$

So using our normalization (A.20), we get

$$\begin{aligned} C_{\beta_3,\beta_2,\beta_1}^{\sigma_2,\sigma_1,\sigma_3} =& \frac{1}{\sqrt{\gamma_0}\Gamma_b(Q)} \left(|S_b(2\beta_1)|^2|S_b(2\beta_2)|^2|S_b(2\beta_3)|^2\right)^{1/4} \sqrt{C(\beta_3,\beta_2,\beta_1)}\begin{Bmatrix} \beta_2 & \beta_1 & \beta_3 \\ \sigma_1 & \sigma_3 & \sigma_2 \end{Bmatrix}_b \\ =& \frac{1}{2^{3/8}\sqrt{\gamma_0}\Gamma_b(Q)} \left(\mu(P_{\beta_1})\mu(P_{\beta_2})\mu(P_{\beta_3})\right)^{1/4} \sqrt{C(\beta_3,\beta_2,\beta_1)}\begin{Bmatrix} \beta_2 & \beta_1 & \beta_3 \\ \sigma_1 & \sigma_3 & \sigma_2 \end{Bmatrix}_b, \end{aligned} \tag{A.26}$$

where all the indices are now invariant under reflection, and this formula again takes exactly the same form as in rational CFTs [3, 53].

## A.4 Racah gauge

Similar to [4], we can also further rescale the three point vertices by a constant(chosen for convenience) times the inverse square root of the bulk structure coefficient and go to the Racah gauge. We have

$$\begin{aligned} F_{\sigma_2,\beta_3}^{\text{Racah}}\begin{pmatrix} \beta_2 & \beta_1 \\ \sigma_3 & \sigma_1 \end{pmatrix} =& \sqrt{\frac{C(\sigma_2,\beta_1,\sigma_1)C(\sigma_3,\beta_2,\sigma_2)}{C(\beta_3,\beta_2,\beta_1)C(\sigma_3,\beta_3,\sigma_1)}} F_{\sigma_2,\beta_3}\begin{pmatrix} \beta_2 & \beta_1 \\ \sigma_3 & \sigma_1 \end{pmatrix} \\ =& \sqrt{|S_b(2\beta_3)|^2|S_b(2\sigma_2)|^2}\begin{Bmatrix} \beta_2 & \beta_1 & \beta_3 \\ \sigma_1 & \sigma_3 & \sigma_2 \end{Bmatrix}_b, \end{aligned} \tag{A.27}$$

and the three point functions become,

$$C_{\beta_3,\beta_2,\beta_1}^{\text{Racah};\sigma_2,\sigma_1,\sigma_3} = \frac{2^{3/8}\sqrt{\gamma_0}\Gamma_b(Q)}{\sqrt{C(\beta_3,\beta_2,\beta_1)}} C_{\beta_3,\beta_2,\beta_1}^{\sigma_2,\sigma_1,\sigma_3} = \left(\mu(P_{\beta_1})\mu(P_{\beta_2})\mu(P_{\beta_3})\right)^{1/4}\begin{Bmatrix} \beta_2 & \beta_1 & \beta_3 \\ \sigma_1 & \sigma_3 & \sigma_2 \end{Bmatrix}_b, \tag{A.28}$$

where the expression for the $6j$ symbol is in (A.12).

The crossing kernel related to the identity module in the usual normalization is given by [57],

$$F_{1,\alpha_k}\begin{pmatrix} \alpha_i & \alpha_j \\ \alpha_i & \alpha_j \end{pmatrix} = C_0(\alpha_i,\alpha_j,\alpha_k)\rho_0(P_k). \tag{A.29}$$

So in Racah gauge, we have the crossing kernel as[20]

$$
\begin{aligned}
F^{\text{Racah}}_{1,\alpha_k}\begin{pmatrix} \alpha_i & \alpha_j \\ \alpha_i & \alpha_j \end{pmatrix} &= C_0(\alpha_i,\alpha_j,\alpha_k)\rho_0(P_k)\sqrt{\frac{C(1,\alpha_j,\alpha_j)C(\alpha_i,\alpha_i,1)}{C(\alpha_i,\alpha_j,\alpha_k)^2}} \\
&= \frac{c_b}{\sqrt{\rho_0(P_i)\rho_0(P_j)\rho_0(P_k)}}\rho_0(P_k)\sqrt{C(1,\alpha_j,\alpha_j)C(\alpha_i,\alpha_i,1)} \\
&= 2\pi c_b \delta(0)\sqrt{\frac{\rho_0(P_k)}{\rho_0(P_i)\rho_0(P_j)}}\,,
\end{aligned}
\tag{A.30}
$$

which also takes the same form as in rational CFTs [153].

### A.5 Further properties of the shrinkable boundary in Liouville theory

Here we show that the shrinkable boundary also satisfy the property that topological defects can also pass through it unobstructed, as in the case of RCFT [2, 153]. The defect in Liouville field theory is discussed in [154, 155]. Defects can fuse with the boundary to generate other boundaries, similar to the case in RCFT. The topological defect in Liouville theory can be written as

$$
\hat{X} = \int_{\frac{Q}{2}+i\mathbb{R}^+} d\alpha \frac{S_{P_x,P_\alpha}[1]}{\mu(P_\alpha)}\mathcal{P}^\alpha\,,
\tag{A.31}
$$

where $\mathcal{P}^\alpha$ is a projection operator, and

$$
S_{P_1,P_2}[1] = 2\sqrt{2}\cos(4\pi P_1 P_2)\,.
\tag{A.32}
$$

Graphically we can represent this operator as

$$
\hat{X} = \int_{\frac{Q}{2}+i\mathbb{R}^+} d\alpha \frac{S_{P_x,P_\alpha}[1]}{\mu(P_\alpha)}\ \Big|^{\alpha}\,.
\tag{A.33}
$$

To prove the topological defects can pass through the boundary freely, we directly verify

$$
\begin{aligned}
\int_{\frac{Q}{2}+i\mathbb{R}^+} d\alpha \frac{S_{P_x,P_\alpha}[1]}{\mu(P_\alpha)}|S_b(2\beta)|^2\ \Big|^{\alpha}\ \bigcirc^{\beta} &= \int_{\frac{Q}{2}+i\mathbb{R}^+} d\alpha \frac{S_{P_x,P_\alpha}[1]}{\mu(P_\alpha)}|S_b(2\beta)|^2 F_{1\gamma}\begin{pmatrix} \alpha & \beta \\ \alpha & \beta \end{pmatrix}\gamma\bigcirc^{\alpha}_{\alpha}\beta \\
&= \int_{\frac{Q}{2}+i\mathbb{R}^+} d\alpha \frac{S_{P_x,P_\alpha}[1]}{\mu(P_\alpha)}|S_b(2\gamma)|^2 F_{1\beta}\begin{pmatrix} \gamma & \alpha \\ \gamma & \alpha \end{pmatrix}\gamma\bigcirc^{\alpha}_{\alpha}\beta \\
&= \int_{\frac{Q}{2}+i\mathbb{R}^+} d\alpha \frac{S_{P_x,P_\alpha}[1]}{\mu(P_\alpha)}|S_b(2\beta)|^2\ \bigcirc^{\beta}\ \Big|^{\alpha}\,,
\end{aligned}
\tag{A.34}
$$

---

[20]The volume divergence is an artifact in Racah gauge as we also divide out an infinite volume factor for the conformal blocks, thus leading to a finite answer when we combine the terms together.

where we have used the following identity and fusion rules twice

$$|S_b(2\beta)|^2 F_{1\gamma}\begin{pmatrix} \alpha & \beta \\ \alpha & \beta \end{pmatrix} = |S_b(2\gamma)|^2 F_{1\beta}\begin{pmatrix} \gamma & \alpha \\ \gamma & \alpha \end{pmatrix}. \tag{A.35}$$

The above identity can be proved via the property (A.29)

$$F_{1\gamma}\begin{pmatrix} \alpha & \beta \\ \alpha & \beta \end{pmatrix} = C_0(\beta, \alpha, \gamma)\rho_0(P_\gamma), \tag{A.36}$$

and $C_0(\alpha, \beta, \gamma)$ is totally symmetric of these three indices. Hence the topological defect line can arbitrarily commute with the boundary, proving that this specific boundary condition is actually the "cloaking boundary condition" that preserves the topological symmetry [2].

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
