# Peer review of "Deriving the non-perturbative gravitational dual of quantum Liouville theory from BCFT operator algebra"

_SciPost Physics, doi:SciPost Phys. 19, 163 (2025)_

## Round 1 · Referee Report · Anonymous (Referee 1) · 2025-10-8

Strengths
1- Every calculation is well-explained and clearly motivated by the text. 2- The figures are clear and help with understanding the concepts. 3- The main message of the paper is emphasized multiple times throughout the paper, so the reader never loses track of the overarching goal of the paper. This helped the paper be very readable!
Weaknesses
1- In section IV, I think the reader would benefit from having more explanation for what the object U(\lambda) is explicitly. It was also not clear to me why this implements a rescaling of the lattice or a depth change, so maybe writing more about it would help the reader understand it. 2- I was a bit confused about the notation of boundary changing operators at the beginning, so it might be beneficial to explain that these are different than the boundary operators in classical BCFT cases, like in the Lewellen papers for example. 3- In the discussion section, it might be useful to illustrate what the 1-4 move does.
Report
Requested changes
1- p.5 third paragraph from the bottom: did you mean figure 5 instead of figure 10? 2- p.5 in figure 3 it is not clear why there is a black tetrahedron drawn in the integral in addition to the blue one. 3- p.14 below the section 5 heading: typo "It is one" --> "One". 4- p.15 top of second paragraph: typo "perculiar" --> "peculiar". 5- p.16 below eqn (A4): 2 typos where two instances of "becomes" --> "become". 6- p.17 in (A10): typo in the 6j curly bracket, probably shouldn't have "an" in its exponent. 7- footnotes on p. 18&19 are formatted in a weird way so it's a bit hard to read. 8- reference [125] has been published now -- 2407.02649.
Recommendation
Publish (easily meets expectations and criteria for this Journal; among top 50%)

---

## Round 1 · Referee Report · Anonymous (Referee 2) · 2025-11-16

Strengths
2-Has explicit formulas and results, especially about the classical limit of 6j symbols.
3-Gives a very concrete realization of tensor network.
4-Has good technically interesting proposals such as the strange correlator and shrinkable boundary conditions.
Weaknesses
2-It seems to be a straightforward generalization of the irrational case, at least conceptually.
Report
They show that after choosing an appropriate “shrinkable boundary” (a continuous analogue of the entanglement brane familiar from RCFTs), the Liouville partition function can be written as a strange correlator pairing. In the semiclassical limit, each such 6j-symbol reduces to the on-shell Einstein–Hilbert action evaluated on a hyperbolic tetrahedron with suitable corner terms. This produces a non-perturbative, discrete formulation of a 3D theory whose large-c behavior matches AdS3 Einstein gravity. The result is an explicit holographic tensor network, which is interesting.
In summary, this is an interesting piece of work. Generalizing from the irrational case allowed the authors to make connections to 3d gravity. The main results appear correct and clearly presented. I recommend it for publication either without or with minor corrections.
Requested changes
1-(optional) The paper would benefit from a fully explicit worked example.
2-(optional) The connections with Teichmüller TQFT and the recently proposed Virasoro TQFT have not been clearly explained (it would strengthen the paper to include a more solid comparison clarifying how the present construction overlaps with or differs from those frameworks).
Recommendation
Publish (easily meets expectations and criteria for this Journal; among top 50%)

---

## Editorial Decision

published